# Genome-wide characterization of 54 urinary metabolites reveals molecular impact of kidney function

Erkka Valo [1,2,3], Anne Richmond [4,5], Stefan Mutter [1,2,3], Emma H. Dahlström [1,2,3], Archie Campbell [4], David J. Porteous [4], James F. Wilson [4,5,6], FinnDiane Study Group*, Per-Henrik Groop [1,2,3,7,8], Caroline Hayward [4,5,9] ✉ & Niina Sandholm [1,2,3,9] ✉

Dissecting the genetic mechanisms underlying urinary metabolite concentrations can provide molecular insights into kidney function and open possibilities for causal assessment of urinary metabolites with risk factors and disease outcomes. Proton nuclear magnetic resonance metabolomics provides a high-throughput means for urinary metabolite profiling, as widely applied for blood biomarker studies. Here we report a genome-wide association study meta-analysed for 3 European cohorts comprising 8,011 individuals, covering both people with type 1 diabetes and general population settings. We identify 54 associations ($p < 9.3 \times 10^{-10}$) for 19 of 54 studied metabolite concentrations. Out of these, 33 were not reported previously for relevant urinary or blood metabolite traits. Subsequent two-sample Mendelian randomization analysis suggests that estimated glomerular filtration rate causally affects 13 urinary metabolite concentrations whereas urinary ethanolamine, an initial precursor for phosphatidylcholine and phosphatidylethanolamine, was associated with higher eGFR lending support for a potential protective role. Our study provides a catalogue of genetic associations for 53 metabolites, enabling further investigation on how urinary metabolites are linked to human health.

Urinary metabolite concentrations are read-outs of biological processes and can inform on the molecular basis of diseases. Automation of metabolomics technologies for urine analyses, such as nuclear magnetic resonance (NMR), has lagged behind blood profiling but now allows for accurate quantification at an entire cohort scale. This may pave the way for wide-spread epidemiological and translational applications analogous to plasma NMR profiling (e.g. in the UK Biobank[1]). Such recent studies have, for example, highlighted 10 urinary metabolites being predictive of diabetic kidney disease (DKD) progression in individuals with type 1 diabetes (T1D)[2] and multiple associations between 49 clinical measures and 12 urinary metabolites in a general population setting[3].

Studying the genetic regulation of urinary metabolites can reveal biological pathways behind the identified biomarkers. Specific to the

[1]Folkhälsan Research Center, Helsinki, Finland. [2]Department of Nephrology, University of Helsinki and Helsinki University Hospital, Helsinki, Finland. [3]Research Program for Clinical and Molecular Metabolism, Faculty of Medicine, University of Helsinki, Helsinki, Finland. [4]Centre for Genomic and Experimental Medicine, Institute of Genetics and Cancer, University of Edinburgh, Western General Hospital, Edinburgh, UK. [5]MRC Human Genetics Unit, Institute of Genetics and Cancer, University of Edinburgh, Western General Hospital, Edinburgh, UK. [6]Centre for Global Health Research, Usher Institute, University of Edinburgh, Edinburgh, UK. [7]Department of Diabetes, Central Clinical School, Monash University, Melbourne, VIC, Australia. [8]Baker Heart and Diabetes Institute, Melbourne, VIC, Australia. [9]These authors contributed equally: Caroline Hayward, Niina Sandholm. *A list of authors and their affiliations appears at the end of the paper. ✉e-mail: caroline.hayward@ed.ac.uk; niina.sandholm@helsinki.fi

urinary biomarkers, is that they can either reflect the systemic (blood) biomarker levels but provide a less tightly regulated and more accessible source of biomarker material compared with blood; or they can reflect changes in the kidney function, related either to changes in the glomerular filtration rate, increased leakage of molecules into the urine, changes in tubular reabsorption into the blood, or originating from the kidney or the urinary system tissue.

Previous research on the genetics of urinary metabolites has identified several hundred loci associated with urinary metabolites[4–7]. Schlosser et al. (2023) identified 622 genomic intervals associated with urinary metabolite concentrations across 1399 metabolites measured in 4912 individuals[7]. Moreover, a study in the UK Biobank identified multiple loci associated with four clinical urinary laboratory measurements in 363,228 individuals[8]. Here, balancing between large sample size and molecular coverage, we have utilized the urinary NMR metabolomics platform detecting 54 urinary metabolites in 8011 individuals to further characterize the genetics of urinary metabolites.

Studying the genetic factors associated with urinary metabolites provides other benefits beyond explaining the underlying biological mechanisms. In particular, if genetic variants can be identified for a urinary metabolite, these variants can serve as genetic instruments in Mendelian randomization (MR) to infer causality between the metabolite and disease outcomes. The identification of multiple robust variants provides more reliable MR results, i.e., by enabling the use of MR methods that account for pleiotropic effects. Although our study includes fewer metabolites than many mass spectrometry based studies, the larger sample size gives us power to potentially identify more associations with urinary metabolites.

In this study, we performed genome-wide association study (GWAS) meta-analyses in a total of 8011 individuals to investigate single nucleotide variants (SNVs) associated with 54 urinary metabolites measured by NMR in one Finnish cohort of individuals with type 1 diabetes (T1D) and two Scottish cohorts from a general population setting. After conditional analysis to identify independent secondary signals, we characterized the identified associations and their molecular basis by analysing the variants' effect on gene expression harnessing relevant expression quantitative trait loci (eQTL) data and performed pathway analyses to obtain wider understanding of the biological pathways affecting each metabolite. Finally, we assessed causal relationships between the metabolites and relevant phenotypes and health outcomes using MR analysis (Fig. 1).

## Results

### Genome-wide association study identified 54 associations with urinary metabolites

We performed GWAS of 54 urinary metabolites in 3 cohorts followed by a meta-analysis (Methods). The analysis included in total of 8011 individuals from the Finnish Diabetic Nephropathy Study (FinnDiane, $n = 3244$, including 1632 men and 1612 women)[9,10], Generation Scotland (GS, $n = 2743$, including 1373 men and 1370 women)[11], and the VIKING study (VIKING, $n = 2024$, including 811 men and 1213 women)[12] (Fig. 1, Supplementary Table 1). We measured 54 urinary metabolites with the Nightingale Health urine NMR platform (Supplementary Data 1 and 2). The metabolites were quantified in absolute concentrations and normalized with urinary creatinine concentration prior to analysis.

Altogether 27 of the 54 metabolites showed significant evidence of heritability ranging from 6% to 36% ($p < 0.05$), with the highest estimates obtained for urinary citrate (36%, $p = 2.3 \times 10^{-20}$), 3-aminoisobutyrate (33%, $p = 1.7 \times 10^{-11}$), and tyrosine (29%, $p = 4.9 \times 10^{-12}$) concentrations (Supplementary Data 3, Supplementary Fig. 1, Methods). Only six metabolites showed evidence of between study heterogeneity in the heritability estimates ($p < 0.05/54 = 9 \times 10^{-4}$), but none of the 27 metabolites with significant heritability estimates, supporting that there are no major differences in the genetic architecture of these metabolites between the studies.

The GWAS meta-analysis results were filtered to variants found in at least 2 out of 3 cohorts and with a minor allele frequency (MAF) ≥ 1%. We identified 26 chromosomal regions harbouring associations with at least one metabolite amongst the 54 metabolites meta-analysed across the three cohorts ($p < 9.3 \times 10^{-10}$; Fig. 2, Methods). In total, the regions contained 34 associations with 19 unique urinary metabolites and three of the 26 regions showed evidence of pleiotropy: the loci on chromosomes 5p15.33 and 17q12 associated with 5 and 4 amino acids respectively, and a locus on chromosome 7p21.1 associated with quinic acid and trigonelline. The GWAS results for glucose were spurious and hard to interpret, and are therefore not reported.

We performed conditional and joint (COJO) multiple-SNV analysis[13] to pinpoint independent signals within these loci and found 54 significant associations ($p < 9.3 \times 10^{-10}$) (Supplementary Fig. 2 and Supplementary Data 4, Methods). In total, 6 metabolites had multiple signals in the same locus, notably, a region on chromosome 5p13.2 in or near the *AGXT2* gene had 15 associations with 3-aminoisobutyrate. Nine of the 54 lead COJO signals showed evidence of between-study heterogeneity. These included 5 secondary signals for the 3-aminoisobutyrate locus on chromosome 5p13.2, with the strongest associations obtained from the FinnDiane study with individuals with T1D. Of note, for four of these secondary signals, the COJO estimate – conditional on other signals within the same locus – was in the opposite direction to the simple single-variant meta-analysis estimate, emphasizing the complexity of this locus with 15 detected independent signals (Supplementary Fig. 3). Furthermore, the glycine association in the *GM2A* gene showed a trend in the opposite direction in the FinnDiane study compared to the two general population studies.

Of the 54 COJO associations, 33 signals were not reported in the GWAS Catalogue for urinary or blood metabolite traits ("novel signals", Table 1), whereas 21 associations were previously reported for relevant urinary and/or blood metabolite traits (Supplementary Table 2). These novel signals included 11 associations with 3-aminoisobutyrate, 4 associations with glycine, 2 associations with 3-hydroxyisovalerate, 4-deoxyeryhronic acid, threonine, and xylose, and finally, single associations with 3-hydroxyisobutyrate, 4-deoxythreonate, citrate, ethanolamine, formate, propylene glycol, quinic acid, trigonelline, tryptophan, and tyrosine (Table 1).

We tested replication of the 33 novel signals for 16 metabolites based on two previous urinary metabolite studies[4,7]. We found evidence of replication ($p < 0.05/24 = 0.0021$ and concordant effect direction) for 16 of the 24 signals that were present in the replication data (Supplementary Data 5). Of note, 9 of the associations had a genome-wide significant p-value ($p < 5 \times 10^{-8}$) for replication, even though they were not reported in the GWAS catalogue; these were either due to i) a more stringent significance threshold required in the original study (e.g., rs2472479 3-hydroxyisobutyrate signal from Schlosser et al. 2023 study) ii) some of the older findings not being reported in the GWAS catalogue (e.g., rs62313082 associated with ethanolamine in Raffler et al. 2015); or iii) many of the secondary independent signals having been filtered out from the GWAS catalogue results (e.g. multiple of the 3-aminoisobutyrate signals on chromosome 5 and replicated in Schlosser et al. 2023).

As we analysed urinary metabolite to creatinine ratios instead of pure metabolite concentrations in order to account for the variation in the urinary concentration and volume, we additionally tested whether the lead metabolite signals would be driven by the denominator (urinary creatinine) instead of the actual metabolite level (nominator). However, none of the COJO lead variants were significantly associated with urinary creatinine ($p > 0.05/54$ for all; Supplementary Fig. 4), supporting that the observed signals are mostly driven by the urinary metabolite levels.

Five of the 54 lead variants were missense variants, two representing previously unknown metabolite associations. rs11567842

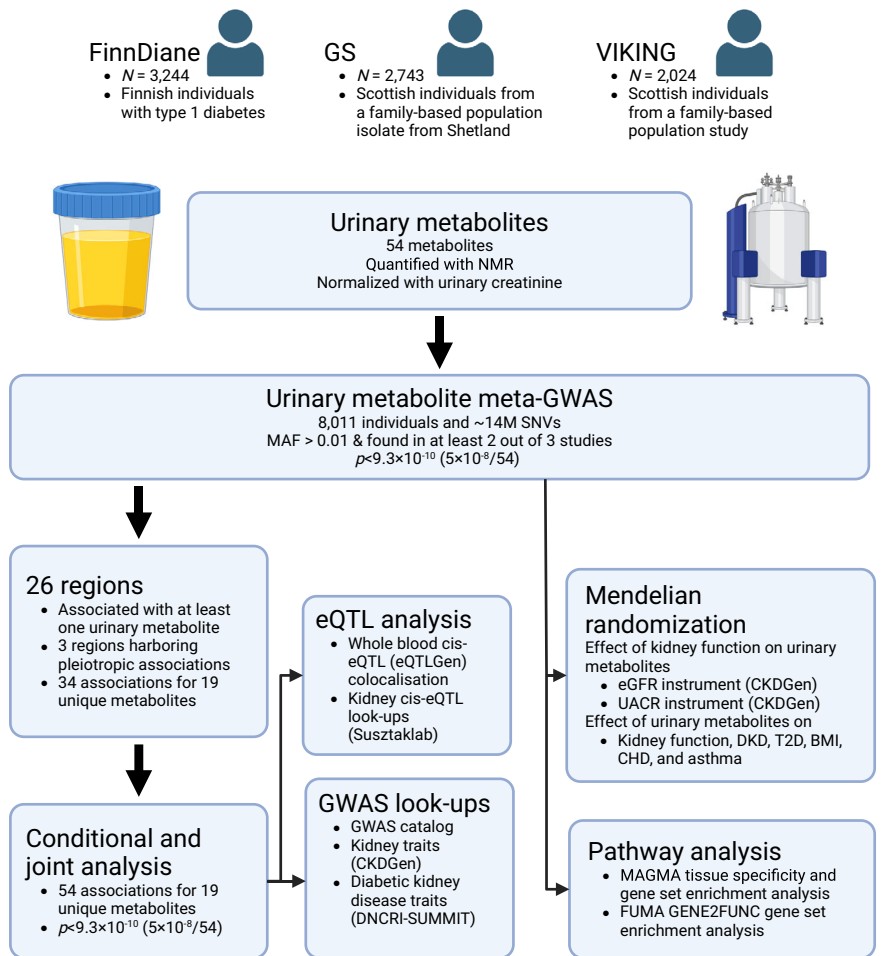

**Fig. 1 | An overview of the genome-wide characterization of the urinary metabolites.** We performed genome-wide association study (GWAS) meta-analyses in a total of 8,011 individuals from three cohorts to investigate genetic variants associated with 54 urinary metabolites measured by NMR. Conditional and joint analysis (COJO) was applied to identify independent secondary signals within the associated genetic loci. The identified associations were characterized by expression quantitative trait loci (eQTL) analysis and association look-ups from GWAS catalogue and kidney GWAS data. We performed two-directional Mendelian Randomization to study the causal associations between the urine metabolites and health outcomes. Pathway analyses were performed to obtain wider understanding of the biological pathways affecting each metabolite. FinnDiane: Finnish Diabetic Nephropathy Study. GS: Generation Scotland. SNV: single nucleotide variant. Meta-GWAS two-sided p-values calculated with METAL applying inverse variance weighted method and genomic control correction for the individual study level results. Conditional and joint analysis two-sided p-values calculated with GCTA-COJO applying conditional and joint analysis of independently associated variants. eGFR: estimated glomerular filtration rate. UACR: urinary albumin creatinine ratio. DKD: diabetic kidney disease. T2D: type 2 diabetes. Created in BioRender. Valo, E. (2024) BioRender.com/a61e932.

(*SLC13A2* p.Ile599Leu) was associated with urinary citrate concentration and has previously been associated with blood urea nitrogen concentration[14]. *SLC13A2* encodes Solute Carrier Family 13 Member 2, which is a kidney sodium-coupled citrate transporter[15]. We have previously shown that urinary citrate concentration is associated with progression of DKD[2]. The citrate-associated rs11567842 was nominally associated with multiple DKD phenotypes ($p = 0.03$-$0.001$)[16], although these genetic associations did not remain after correction for multiple testing. At a locus on chromosome 5p13.2, including 15 independent signals associated with 3-aminoisobutyrate, three of the lead variants were missense variants but in three different genes: rs37369 (*AGXT2* p.Val140Ile), rs2308957 (*RAD1* p.Gly114Asp), and rs138373837 (*NADK2* p.Arg211His). The rs37369 (*AGXT2* p.Val140Ile) variant has previously been associated with urinary and plasma 3-aminoisobutyrate levels[17,18] whereas rs2308957 (*RAD1* p.Gly114Asp) and rs138373837 (*NADK2* p.Arg211His) had not. Other lead variants in the region were eQTLs for *AGXT2* or both *AGXT2* and *RAD1* in the kidney. *AGXT2* encodes a mitochondrial alanine–glyoxylate aminotransferase 2, expressed particularly in kidney and liver in the human protein atlas, and is the biologically most plausible gene underlying the association signal, as it

converts (R)-3-aminoisobutyric acid and pyruvate to 2-methyl-3-oxopropanoate and alanine (Reactome ID: R-HSA-909780.2). In single-cell RNAseq of human kidneys, the gene is expressed specifically in the proximal convoluted tubules (Supplementary Fig. 5)[19]. Finally, rs1047891 (*CPS1* p.Thr1406Asn) has previously been associated with 266 traits including the currently observed glycine association in plasma[20], as well as with estimated glomerular filtration rate (eGFR)[21].

Of note, all the identified missense variants were predicted to be tolerated or benign by SIFT and PolyPhen.2 algorithms, but they may be sufficient to cause subtle changes in the protein function seen as altered urinary metabolite concentrations.

## eQTL data in kidney and whole blood highlights membrane transport proteins

Considering that most of the identified variants were non-coding, we next used available gene expression quantitative trait loci (eQTLs) data to help assess whether the identified variants influence the expression of nearby genes (Methods). As the kidneys play an important role in urine production and filtration, we first investigated the regulatory effect of the identified variants on gene expression in kidneys. We did

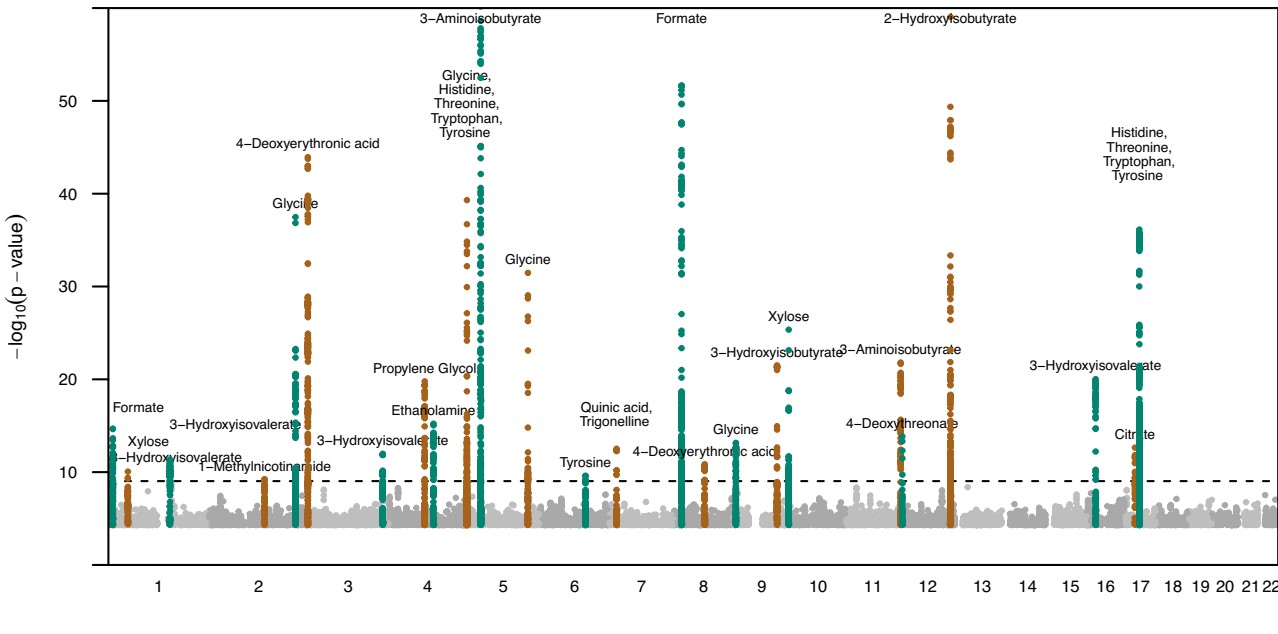

**Fig. 2 | Manhattan plot of signals with $p < 5.0×10^{-5}$ for the metabolites.** Signals from different metabolites are clumped together if they are within 50 kb from another signal. Pruned genome-wide significant signals with $p < 5 × 10^{-8}/54 = 9.3 × 10^{-10}$ and variants 1 Mb around them are highlighted with two alternating colours.

Note: y-axis clipped at 60. P-values calculated with METAL applying inverse variance weighted method and genomic control correction for the individual study level results. All p-values are two-sided.

this by querying our lead variants in Human Kidney eQTL Atlas[22], containing kidney eQTL data from 686 micro-dissected human kidney samples (whole kidney, and dissected into tubular and glomerular tissues). Half of the identified variants for urinary metabolites (26 of 54) were *cis*-eQTLs, i.e., associated with gene expression of a nearby gene (1 Mb), i.e., an eGene, in either kidney tubules, glomeruli, and/or whole kidney ($p < 5.3 × 10^{-4}$, Table 2, Supplementary Data 6) influencing the expression of 30 genes. The kidney eQTL data provided a likely candidate, for e.g., for rs62313082 (upstream of the *RCC2P8* gene) associated with urinary ethanolamine. This variant was a kidney eQTL for *ETNPPL* ($p = 1.2 × 10^{-34}$, Table 2), which catalyses breakdown of phosphoethanolamine[23], and thus, represents a plausible gene underlying the metabolite association.

We further extended the eQTL look-ups to eQTLGen whole blood data[24], allowing us to also investigate the colocalization of the eQTL association with the metabolite association (Methods). Although the eQTLGen data may miss eQTLs for genes expressed only in specific tissues, such as kidneys, the larger sample size of eQTLgen ($n = 31,684$) also enables the detection of weaker signals for general eQTL associations. In total, 25 of the 54 lead variants that associated with the urinary metabolites were cis-eQTLs influencing the expression of 106 nearby genes in whole blood ($p < 5.5 ×10^{-5}$; Supplementary Data 6). Eleven of these blood eQTLs were also kidney eQTLs for the same genes; e.g. the 3-hydroxyisobutyrate- associated variant (rs2472479) located upstream of *NIPSNAP3B* was both a kidney eQTL and a blood eQTL for *NIPSNAP3A*, encoding NipSnap homolog proteins 3B and 3A with putative roles in vesicular transport[25]. However, we found that only one third of the eQTLs associations colocalized with our urinary metabolite-associations (34 of 106, posterior probability (PP) > 0.5); 24 of these resulted from eQTL signals for 6 genes colocalizing with 4 amino acid signals in a single region on chromosome 17q12 (Table 2).

Altogether nine eQTLs that colocalized with the metabolite-association were not detected in the kidney eQTL datasets. These included the variant rs62565993, a strong eQTL for *GLDC* in whole blood ($p = 2.4×10^{-79}$), colocalizing with the glycine-association (PP = 0.98).

Given that *GLDC* encodes glycine decarboxylase, a component of the glycine cleavage system catalysing the degradation of glycine[26], it is a plausible underlying gene for the urinary glycine association. However, for the 1-Methylnicotinamide-associated variant rs17322446, the closest gene (*ACMSD*) may be more plausible due to shared pathways with 1-Methylnicotinamide[27], although kidney eQTL and blood eQTL colocalizations indicated other genes (Table 2).

Many of the eQTL-associated genes, particularly in kidneys, were members of the SLC (Solute Carrier) family, which encode molecules facilitating the transport of metabolites across cell membranes (Table 2). For example, rs10788884, which represents a new association for xylose, was a strong kidney eQTL for *SLC5A9* ($p = 10^{-52}$), encoding a sodium-dependent glucose transporter (SGLT4) expressed in the intestine and the kidneys[28]. Although it primarily functions in the transport of mannose, 1,5-anhydro-D-glucitol, and fructose[28] it is plausible that it could also be involved in the transport of other sugars like xylose. In addition, two previously unreported urinary tyrosine- and tryptophan-associated variants (rs7704882 and rs7704058, in full LD in the European population: $r^2 = 1$) were strong kidney eQTLs (Table 2) for *SLC6A18*. However, the association signal for urinary tyrosine around rs7704882 showed evidence of colocalization with the *SLC6A19* kidney eQTL signal (Fig. 3 and Supplementary Fig. 6), which indeed encodes a known neutral amino acid transporter for e.g., tyrosine, whereas the nearby paralogue *SLC6A18* seems to be more specific for glycine[29]. Of note, tyrosine was one of the amino acids associated with progression of DKD to kidney failure in our previous observational study[2]. *SLC19A19* was also indicated as the target eGene for the already reported metabolite[4,6,30] and eGFR-associated[31] variant, rs11133665 (Table 3), located 37 kb away. The other solute carrier genes revealed by the eQTL data were the *SLC16A10* gene (expression affected by urinary tyrosine-associated rs241768) encoding a known transporter of tyrosine and other metabolites[32]. The urinary 3-aminoisobutyrate-associated rs2080403 was a kidney and blood eQTL for *SLC6A13*, which it also colocalized with (PP = 0.82; Table 2; Supplementary Fig. 7) suggesting 3-aminoisobutyrate as an unknown substrate of *SLC6A13* as previously hypothesized[4,33]. In addition, the

**Table 1 | Variants associated with urinary metabolites (two-sided $p < 9.3 \times 10^{-10}$) with no previously reported associations in the GWAS catalogue with the same metabolite in blood or urine (window size = -/+500 kb, $r^2 > 0.8$, and $p < 5 \times 10^{-8}$)**

| CHR:POS:EA:NEA | Rsid | Gene | Variant type | Metabolite | EAF | Beta (SE) | P | N | Prev. Associations |
|---|---|---|---|---|---|---|---|---|---|
| 1:6334301:A:G | rs114200864 | ACOT7 | intron | 3-Hydroxyisovalerate | 0.041 | −0.28 (0.05) | $2.9 \times 10^{-10}$ | 8007 | No |
| 1:11940483:T:C | rs4846068 | SBF1P2 / 1p36.22 | downstream (0.02 kb) | Formate | 0.585 | 0.13 (0.02) | $2.2 \times 10^{-15}$ | 7998 | Yes |
| 1:48690229:A:C | rs10788884 | SLC5A9 | intron | Xylose | 0.674 | 0.11 (0.02) | $1.1 \times 10^{-10}$ | 7270 | Yes |
| 2:241813788:T:C | rs10933641 | AGXT | intron | 4-Deoxyerythronic acid | 0.277 | 0.19 (0.02) | $2.6 \times 10^{-20}$ | 7609 | Yes* |
| 3:182758040:T:C | rs4859267 | MCCC1 | intron | 3-Hydroxyisovalerate | 0.294 | 0.13 (0.02) | $1.3 \times 10^{-12}$ | 8007 | Yes |
| 4:88213884:T:C | rs6811902 | MIR570S / 4q22.1 | downstream (8 kb) | Propylene Glycol | 0.601 | −0.16 (0.02) | $2.8 \times 10^{-20}$ | 5039 | Yes |
| 4:109716840:A:T | rs62313082 | RCC2P8 / 4q25 | upstream (6 kb) | Ethanolamine | 0.379 | 0.14 (0.02) | $7.7 \times 10^{-16}$ | 6885 | No |
| 5:1188285:A:G | rs11133665 | TERLR1 / 5p15.33 | upstream (10 kb) | Glycine | 0.261 | −0.13 (0.02) | $1.6 \times 10^{-13}$ | 7795 | Yes |
| 5:1188285:A:G | rs11133665 | TERLR1 / 5p15.33 | upstream (10 kb) | Threonine | 0.259 | −0.15 (0.02) | $1.3 \times 10^{-16}$ | 7960 | Yes |
| 5:1225434:T:C | rs7704882 | SLC6A18 / 5p15.33 | upstream (0.06 kb) | Tyrosine | 0.794 | −0.20 (0.02) | $4.6 \times 10^{-24}$ | 7833 | No* |
| 5:1225613:A:G | rs7704058 | SLC6A18 | synonymous | Tryptophan | 0.796 | −0.15 (0.02) | $5.1 \times 10^{-14}$ | 7716 | No* |
| 5:34742391:C:G | rs9292560 | RAI14 | intron | 3-Aminoisobutyrate | 0.633 | 0.13 (0.02) | $2.3 \times 10^{-12}$ | 6347 | No* |
| 5:34849791:A:T | rs336494 | TTC23L | intron | 3-Aminoisobutyrate | 0.144 | −0.19 (0.03) | $4.2 \times 10^{-12}$ | 6347 | No* |
| 5:34868497:A:T | rs72732827 | TTC23L | 3'-UTR | 3-Aminoisobutyrate | 0.987 | −0.60 (0.08) | $1.2 \times 10^{-13}$ | 6347 | No* |
| 5:34896132:A:G | rs138425947 | TTC23L | intron | 3-Aminoisobutyrate | 0.978 | −0.51 (0.07) | $5.4 \times 10^{-14}$ | 6347 | No* |
| 5:34899723:T:C | rs56007938 | TTC23L | 3'-UTR | 3-Aminoisobutyrate | 0.015 | 0.66 (0.08) | $1.1 \times 10^{-17}$ | 6347 | No* |
| 5:34911884:T:C | rs2308957 | RAD1 | missense: p.Gly114Asp | 3-Aminoisobutyrate | 0.015 | −0.78 (0.09) | $3.2 \times 10^{-18}$ | 6347 | No* |
| 5:34982167:T:C | rs116116288 | AGXT2 / 5p13.2 | downstream (20 kb) | 3-Aminoisobutyrate | 0.012 | 0.88 (0.09) | $8.2 \times 10^{-25}$ | 6347 | No* |
| 5:35000653:T:C | rs7737763 | AGXT2 | intron | 3-Aminoisobutyrate | 0.434 | −0.44 (0.03) | $2.5 \times 10^{-67}$ | 6347 | No* |
| 5:35045127:A:G | rs187490 | AGXT2 | intron | 3-Aminoisobutyrate | 0.675 | 0.73 (0.04) | $2.0 \times 10^{-60}$ | 6347 | No* |
| 5:35053452:A:G | rs185217270 | PRLR | intron | 3-Aminoisobutyrate | 0.980 | −0.80 (0.09) | $2.0 \times 10^{-17}$ | 6347 | No* |
| 5:36219710:T:C | rs138373837 | NADK2 | Missense: p.Arg211His | 3-Aminoisobutyrate | 0.019 | −0.46 (0.07) | $2.0 \times 10^{-12}$ | 6347 | Yes* |
| 5:150624099:T:C | rs72794144 | GM2A | intron | Glycine | 0.028 | 0.38 (0.06) | $2.5 \times 10^{-11}$ | 7795 | No* |
| 5:150702299:A:G | rs61067578 | SLC36A2 | intron | Glycine | 0.842 | 0.24 (0.02) | $1.1 \times 10^{-24}$ | 7795 | Yes* |
| 7:17287998:A:G | rs2106727 | AHR | intron | Quinic acid | 0.353 | −0.12 (0.02) | $3.1 \times 10^{-13}$ | 7657 | Yes |
| 7:17287998:A:G | rs2106727 | AHR | intron | Trigonelline | 0.355 | −0.10 (0.02) | $6.5 \times 10^{-11}$ | 8011 | Yes |
| 8:74868909:A:G | rs2661850 | ELOC | intron | 4-Deoxyerythronic acid | 0.691 | −0.12 (0.02) | $1.5 \times 10^{-11}$ | 7609 | Yes |
| 9:6649491:T:C | rs62565993 | GLDC / 9p24.1 | upstream (4 kb) | Glycine | 0.124 | 0.20 (0.03) | $3.4 \times 10^{-14}$ | 7795 | No |
| 9:107525165:T:G | rs2472479 | NIPSNAP3B / 9q31.1 | upstream (1 kb) | 3-Hydroxyisobutyrate | 0.565 | −0.16 (0.02) | $5.0 \times 10^{-22}$ | 7850 | No |
| 9:136146597:T:C | rs550057 | ABO | intron | Xylose | 0.244 | 0.20 (0.02) | $8.9 \times 10^{-26}$ | 7270 | Yes |
| 12:4521511:A:T | rs78470967 | FGF6 / 12p13.32 | downstream (20 kb) | 4-Deoxythreonate | 0.043 | 0.34 (0.04) | $2.0 \times 10^{-14}$ | 7612 | Yes |
| 17:26824156:A:G | rs15567842 | SLC13A2 | missense: p.Ile599Leu | Citrate | 0.642 | −0.11 (0.02) | $3.0 \times 10^{-13}$ | 7773 | Yes |
| 17:37631883:C:G | rs11078902 | CDK12 | intron | Threonine | 0.241 | −0.12 (0.02) | $3.7 \times 10^{-10}$ | 7960 | Yes |

CHR:POS:EA:NEA Chromosome position (GRCh37), effect allele, and non-effect allele. Rsid variant rs-identifier. Gene Closest gene. Variant type: consequence of the variant on the protein sequence. Closest genes and variant types found using Ensembl VEP (GRCh38 v110). Metabolite the associated urinary metabolite. EAF effect allele frequency. Beta (SE) effect estimate for the effect allele (effect estimate standard deviation). P: p-value of the association from the COJO conditional and joint analysis of independently associated variants. N: number of individuals in the analysis. Prev. Associations: previous associations found in GWAS catalog.
*independent signal in a previously reported locus for the same metabolite.

three variants that associated with glycine on chromosome 5q33.1 included two intronic variants, rs61067578 and rs147000073, in the *SLC36A2* gene encoding a proton-coupled amino acid transporter involved in the reabsorption of small amino acids such as glycine, proline, and alanine in the proximal tubules of the kidneys[34].

## Gene set, pathway and tissue enrichment analyses

To gain insight into the relevant tissues and molecular pathways underlying the urinary metabolite concentrations, we performed two different types of gene set enrichment analyses (Methods). As a first approach, we used MAGMA gene set analysis that first annotates all variants, without any *p*-value threshold, to the underlying or flanking genes and evaluates the gene-level significance. MAGMA tissue expression analysis identified a positive relationship between the highly expressed genes in adipose tissue and cis-Aconitate genetic associations ($p = 6.9 \times 10^{-5}$); as well as between kidney and glycine ($p = 1.5 \times 10^{-4}$) and pituitary gland and pyroglutamate (Supplementary Table 3); confirmatory with some prior studies.

MAGMA gene set enrichment analysis identified nine significantly enriched gene sets ($p < 3 \times 10^{-6}$, Supplementary Data 7). After the strongest enrichment between tyrosine and the positional chr5p15 breast cancer locus, the second strongest enrichment was obtained between threonine and "tachykinin receptors bind tachykinins" pathway ($p = 1.1 \times 10^{-7}$). Of note, the five tachykinin and their receptor genes are all located in different chromosomes, thus representing a true genome-wide enrichment. Tachykinins are neuropeptides derived from alternate processing of the three tachykinin genes. They are expressed throughout the nervous and immunological system, participate in a variety of physiological processes, and contribute to multiple disease processes, including acute and chronic inflammation and pain, fibrosis, affective and addictive disorders, functional disorders of the intestine and urinary bladder, infection, and cancer[35]. Other significant gene sets included enrichment between 4-deoxyerythronic acid and pyruvate family amino acid metabolic process genes, 3-hydroxyisovalerate and uronic acid metabolic process genes, glycolic acid and eukaryotic translation initiation factor 3 complex proteins, tryptophan and genes involved in APC/C:Cdc20 mediated degradation of Cyclin B, and 4-deoxythreonate and genes involved in neuron intrinsic apoptotic signaling pathway in response to oxidative stress.

While MAGMA uses a straightforward approach to annotate the associated variants to the underlying or flanking genes, the gene affected by the GWAS signal is not always the closest one, and we therefore also utilized the FUMA gene set enrichment analysis as a complementary approach. We included only variants reaching a $p < 1 \times 10^{-5}$, but utilised eQTL associations in addition to the positional mapping of variants to genes. Altogether 26 of the 54 metabolites were found to have significant ($p < 0.05$) associations with gene set pathways following FUMA analysis (Supplementary Data 8). The metabolites 3-aminoisobutyrate, histidine, threonine, tryptophan, tyrosine and valine were associated with eGFR, and 4-deoxyerythronic acid was associated with urate concentrations, suggesting a role in kidney health.

Breast cancer was found to be most abundant with related pathways significantly associated with 12 of the metabolites, followed by asthma-related pathways which were significantly associated with 11 metabolites. Of note, the cancer gene sets typically represent single chromosomal loci with a gene cluster, rather than genome-wide enrichment. The amino acid biomarkers histidine, threonine, tryptophan, tyrosine, and valine showed very similar results being significantly associated with the same pathways. These include inflammatory bowel disease, atrial fibrillation, rheumatoid arthritis, menopause, systemic lupus erythematosus and polycystic ovary syndrome. This is due to a GWAS hit on chromosome 17q12, which is present in all 5 amino acids and thus driving most of the associations with the gene set pathways.

## Kidney health causally affects urinary metabolites

As urinary metabolites may reflect kidney health, we investigated whether the identified variants are also associated with kidney disease traits in the general population (CKDGen meta-analysis)[31] and in individuals with diabetes (DNCRI-SUMMIT meta-analysis)[16] (Methods). Indeed, seven of the variants were genome-wide significantly ($p < 5 \times 10^{-8}$) associated with eGFR, a main measure to monitor kidney health, and 4 variants with chronic kidney disease (CKD) ($p < 3.6 \times 10^{-4}$) in the general population. Four variants were also nominally ($p < 0.05$) associated with DKD or kidney failure in diabetes (Table 3).

As multiple urinary metabolite-associated variants were also associated with kidney disease traits we tested whether kidney health causally affects urinary metabolite concentrations. We performed two-sample Mendelian randomization analysis using two kidney function markers, eGFR and urinary albumin-creatinine ratio (UACR), as the exposures, and metabolite concentrations as the outcomes (Supplementary Table 4, Methods). MR assumes that the genetic variants (instrumental variables) are associated with the exposure, are not associated with any confounders of the exposure-outcome relationship and impact the outcome only through the exposure and not via independent pleiotropic pathways. To avoid violation of the relevance assumption underlying MR, we used a genetic instrument for eGFR, composed of 150 independent ($r^2 = 0.001$) genome-wide significant SNVs identified in the CKDGen GWAS meta-analysis[31]. Genetically instrumented eGFR was associated ($p < 4.7 \times 10^{-4}$) with the urinary metabolite concentrations of 4 amino acids (alanine, glutamine, leucine, and valine), as well as 9 other metabolites: 2-hydroxyisobutyrate, 3-hydroxyisovalerate, ethanolamine, formate, glycine, glycolic acid, pseudouridine, pyroglutamate, and uracil (Supplementary Data 9, Supplementary Fig. 8). This suggests a causal association of glomerular filtration rate on these urinary metabolites. For all the metabolites, higher eGFR was associated with higher metabolite concentration in the urine. Next, we used complementary MR methods (median- and mode-based methods as well as MR Egger), with differing underlying assumptions, to assess the robustness of our findings with respect to pleiotropy. The causal estimates were directionally consistent across different MR analysis methods, suggesting no significant pleiotropy, for most of the metabolites. Moderate heterogeneity between the variant-specific causal estimates (30-32%, Supplementary Data 9), were detected for valine and alanine. However, the Egger intercept did not significantly deviate from zero ($p > 0.05$, Supplementary Data 9). Furthermore, eGFR remained associated ($p < 0.05$) with glycolic acid, 3-hydroxyisovalerate, and pseudouridine even with the MR Egger method which is more robust against pleiotropic effects. Finally, as some genetic loci associated with eGFR may reflect creatinine metabolism rather than kidney function, we performed sensitivity analyses by filtering the eGFR loci to those additionally associated ($p < 0.05$) with two alternative measurements of kidney function, namely cystatine C based eGFR[21], and blood urea nitrogen[31]; all detected eGFR-to-metabolite associations remained significant in one or both of these filtered analyses, supporting that the observed causal pathways belong to the kidney function rather than relating to the creatinine metabolism (Supplementary Data 10).

As eGFR causally affected multiple metabolites we tested whether some of the metabolite GWAS associations might be driven by association with eGFR. However, adjusting the analysis with eGFR in the FinnDiane study did not markedly affect the effect size estimates for the 53 COJO lead variants included in the FinnDiane GWAS data set (Supplementary Fig. 9A).

## Urinary metabolites potentially causally linked to kidney function and body mass index

We also performed two-sample Mendelian Randomization to test whether urinary metabolites are causal risk factors or reflect causal biological processes leading to CKD and other chronic diseases

**Table 2 | Expression quantitative trait loci (eQTL, two-sided $p < 5.3 \times 10^{-4}$) target genes in kidney (K), glomeruli (G) and tubule (T) for the COJO lead signals, and target genes of whole blood eQTL signals colocalizing (PP > 0.5) with COJO lead signals**

| Chr:pos | Rsid | Metabolite | Closest gene (other relevant) | Kidney eQTL gene (tissue) | Colocalization (PP) |
|---|---|---|---|---|---|
| 1:11940483 | rs4846068 | Formate | SBF1P2 (MTHFR[61]) | PLOD1 (K[p], Blood) | |
| 1:48690229 | rs10788884 | Xylose | SLC5A9[28] | SLC5A9 (K, T, G) | |
| 1:151904146 | rs2999545 | 3-Hydroxyisovalerate | KRT8P28 (THEM4/5*) | THEM4 (K, T, G, Blood), S100A10 (K[p], G, Blood) | THEM4 (0.97) |
| 2:135598913 | rs17322446 | 1-Methylnicotinamide | ACMSD[27] | TMEM163 (K, T, G, Blood), AC016725.4 (G, Blood) | AC016725.4 (0.84)CCNT2 (0.54) |
| 2:241793545 | rs55649245 | 4-Deoxyerythronic acid | AGXT* | AGXT (G) | |
| 2:241813788 | rs10933641 | 4-Deoxyerythronic acid | AGXT* | MAB21L4 (K, T, G) AGXT (K, T, G) | |
| 3:182758040 | rs4859267 | 3-Hydroxyisovalerate | MCCC1* | MCCC1 (K[p], Blood) | MCCC1-AS1 (0.7) |
| 4:88213884 | rs6811902 | Propylene Glycol | MIR5705 | HSD17B11 (K, T, G[p], Blood) | |
| 4:109716840 | rs62313082 | Ethanolamine | RCC2P8 (ETNPPL[23]) | ETNPPL (K, T, G) | |
| 5:1188285 | rs11133665 | Glycine, Histidine, Threonine, Tryptophan, Tyrosine | TERLR1 (SLC6A19[4,6,30]) | SLC6A19 (K, T, G) | |
| 5:1225434 | rs7704882 | Tyrosine | SLC6A18* | SLC6A18 ((K, T, G) | |
| 5:1225613 | rs7704058 | Tryptophan | SLC6A18* | SLC6A18 (K, T, G) | |
| 5:34993215 | rs11744796 | 3-Aminoisobutyrate | AGXT2* | AGXT2 (K, T, G) DNAJC21 (K[p], T[p], G, Blood) RAD1 (K) | |
| 5:35039437 | rs2279651 | 3-Aminoisobutyrate | AGXT2* | AGXT2 (K, T) | |
| 5:35045127 | rs187490 | 3-Aminoisobutyrate | AGXT2* | AGXT2 (K) | |
| 6:111492119 | rs241768 | Tyrosine | SLC16A10[33] | SLC16A10 (K, T) | |
| 9:6649491 | rs62565993 | Glycine | GLDC[26] | | GLDC (0.98) |
| 9:107525165 | rs2472479 | 3-Hydroxyisobutyrate | NIPSNAP3B[25] | NIPSNAP3A (K, T, G, Blood) | |
| 9:136146597 | rs550057 | Xylose | ABO | ABO (K. Blood) | ABO (0.87) |
| 12:345369 | rs2080403 | 3-Aminoisobutyrate | SLC6A13[4,33] | SLC6A13 (K, T, G), CCDC77 (K, T, Blood), AC007406.2 (K), RP11-283I3.4 (T) | SLC6A13 (0.82) NINJ2 (0.61) |
| 12:122344302 | rs1795967 | 2-Hydroxyisobutyrate | PSMD9 (HPD[62]) | CFAP251 (K), WDR66 (T[p]) | RSRC2 (0.64) |
| 16:20557634 | rs7499358 | 3-Hydroxyisovalerate | ACSM2B* (ACSM1/2 A/2B/3/5*) | ACSM2B (G, T), ACSM1 (T[p], Blood) | |
| 16:20608891 | rs540815683 | 3-Hydroxyisovalerate | ACSM5P1 (ACSM1/2 A/2B/3/5*) | | U6 (0.5) |
| 17:37631883 | rs11078902 | Threonine | CDK12 | PGAP3 (K, T, G, Blood), FBXL20 (K, T, G[p], Blood), MED1 (K, T[p], Blood), RP11-690G19.3 (T) | FBXL20 (0.85), MED1 (0.81)CTB-131K11.1 (0.79), NR1D1 (0.77), PSMD3 (0.72), PCGF2 (0.61) |
| 17:37633970 | rs12453397 | Tryptophan, Tyrosine | CDK12 | PGAP3 (K, T, G, Blood),FBXL20 (K, T, G[p], Blood)MED1 (K, T, Blood), RP11-690G19.3 (T) | MED1 (0.86, 0.86)**, NR1D1 (0.77, 0.78)**, PSMD3 (0.76, 0.72)**, FBXL20 (0.76, 0.8)** CTB-131K11.1 (0.75, 0.78)**, PCGF2 (0.68, 0.64)** |
| 17:37636695 | rs4795371 | Histidine | CDK12 | PGAP3 (K, T, G, Blood), FBXL20 (K, T, G[p], Blood), MED1 (K, T[p], Blood), RP11-690G19.3 (T) | CTB-131K11.1 (0.85), MED1 (0.82), NR1D1 (0.79), FBXL20 (0.77), PSMD3 (0.76), PCGF2 (0.67) |

Closest gene (other relevant): closest genes were queried from Ensembl VEP (GRCh38 v110); other relevant nearby genes (+/-500 kbp) were sought from the literature and queried from the EGEA data base (https://github.com/fauman/EGEA). Genes found in the EGEA database are denoted by an asterisk (*) or with the number referring to the relevant citation. The exact p-values for the association of the target gene with an eQTL are presented in Supplementary Data 6. The label 'Blood' indicates that the kidney eQTL target gene is also a target gene in blood. The letter [p] indicates that the eQTL is a proxy variant of the lead signal with $r^2 > 0.8$. PP refers to the posterior probability of colocalization calculated with a Bayesian test for colocalization[55]. A double asterisk (**) denotes the posterior probabilities for tryptophan and tyrosine, respectively.

(Supplementary Table 4, Methods). The analysis suggested that higher urinary 3-hydroxyhippurate, quinic acid and trigonelline concentrations are causally associated with higher body mass index (BMI), higher eGFR (i.e., reflecting better kidney health), and contradictorily, also with higher UACR (i.e., reflecting worse kidney health; Table 4). All these three metabolites are found in coffee; their instrumental variables (IVs) rs2106727 and rs6968554 are located in the *AHR* locus, and are in strong LD with rs4410790 that has been previously associated with caffeine intake ($p = 2.0 \times 10^{-249}$)[36], suggesting that the metabolite

associations observed with Mendelian Randomization may reflect the underlying association with coffee consumption. Indeed, in line with our observation of the three metabolites associated with higher eGFR, a previous Mendelian Randomization study suggested that coffee consumption has a beneficial effect on kidney function and albuminuria[37]. Why the three urinary metabolites were simultaneously associated with higher UACR in our data remains unclear. For BMI, previous MR studies have found contradictory evidence regarding the causality between coffee consumption and BMI or obesity[38,39]. One

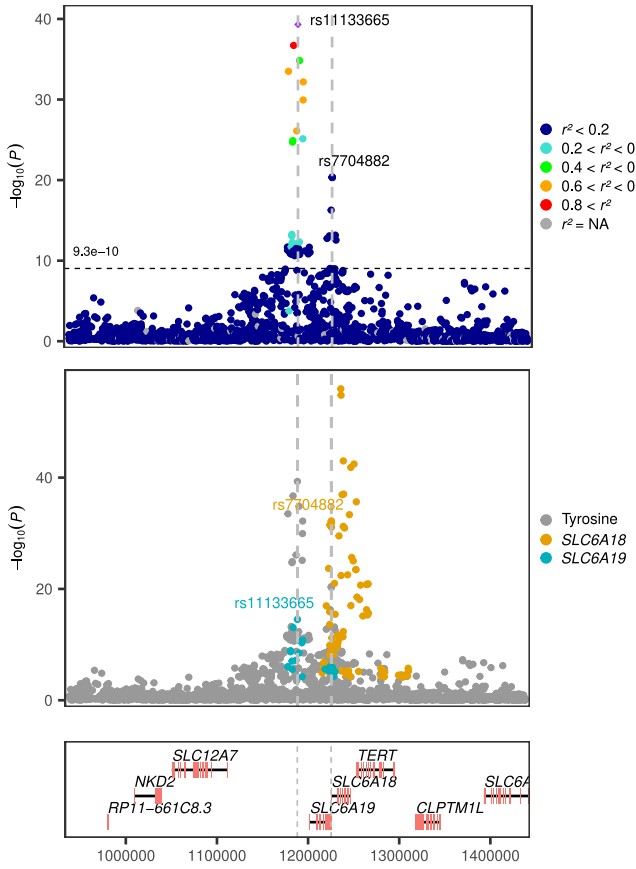

**Fig. 3 | Regional association with tyrosine for lead variants rs11133665 and rs7704882 on chromosome 5.** Upper panel shows LocusZoom plot centred around the previously known rs11133665 variant, and the previously unreported signal at rs7704882 independently associated with tyrosine. P-values were calculated with METAL applying inverse variance weighted method and genomic control correction for the individual study level results. The middle panel shows kidney eQTL associations for *SLC6A18* and *SLC6A19* overlaid on top of the tyrosine association signals, highlighting lead variants rs11133665 (eQTL for *SLC6A19* in kidney) and rs7704882 (eQTL for *SLC6A18* in the kidney)[22]. The lower panel contains the protein-coding genes in the region with exons highlighted. The $r^2$ values with rs11133665 are calculated based on 1000 Genomes phase 3 European population. Variants with no $r^2$ information are not shown. P-values from two-sided tests.

challenge may be the reverse causality; in our study, BMI was not causally affecting the three metabolites, or any other urinary metabolite (Supplementary Data 11). In general, the urinary metabolites may provide a more exact estimate of the coffee intake than self-reported data on coffee consumption, and thus, our findings add to the previous evidence of a BMI-increasing effect on coffee consumption. However, we note that the *AHR* variants rs2106727 and rs6968554 are also associated with other traits such as blood lipid concentrations in the GWAS catalog ($p < 5 \times 10^{-8}$), indicating potential pleiotropic effects.

In addition, the genetic instrument for urinary ethanolamine, was associated with higher eGFR ($p = 6.1 \times 10^{-8}$); of note, this association was not evident when using cystatin C based eGFR ($p = 0.016$) and BUN ($p = 0.18$) as alternative outcome measurements of kidney function (Supplementary Data 12). The genetic instrument was based on the rs62313082 variant which is also a kidney eQTL for the *ETNPPL* gene (Tables 2 and 4) but has not been associated with other traits in the GWAS catalogue (Supplementary Table 2). Moreover, the MR analysis suggested that the genetic instruments for urinary 1-methylnicotinamide ($p = 2.3 \times 10^{-5}$) and 4-deoxythreonate ($p = 7.6 \times 10^{-5}$) were associated with higher body mass index (BMI;

Table 4). These were genetically instrumented by rs78470967 and rs181558 (4-deoxythreonate), and rs17322446 (1-methylnicotinamide), for which we found a few reported associations in the GWAS catalogue (rs17322446; Quinolinate levels, walking pace and FEV1, rs181558; type 2 diabetes and height). Of note, as these MR findings were based on only one or two genetic variants, complementary MR methods more robust to pleiotropy could not be implemented.

## Discussion

To our knowledge this meta-analysis of three large cohorts represents the largest GWAS on urinary NMR metabolomics to date, enabling us to detect previously unidentified associations with urinary metabolites. We identified 54 genetic associations with urinary metabolites, of which 33 were previously unreported in the GWAS catalogue. In line with the notion that GWAS findings for complex diseases are enriched for regulatory variants[40], many of the metabolite associations were outside genes, but were strongly associated with gene expression in the whole kidney, tubules and glomeruli, for example, solute carriers *SLC6A18* and *SLC6A19*. While it is overall not surprising to find kidney associations for urinary metabolites, our findings may help to further describe how the kidneys regulate systemic metabolism by filtration and reabsorption.

Additionally, we observed 4 amino acids and 9 additional metabolites whose urinary concentrations were causally influenced by the glomerular filtration rate in the kidneys. This finding suggests that the clinical studies investigating these metabolites as potential biomarkers of disease risk should include the glomerular filtration rate as a covariate in the analysis. In the opposite direction, MR analysis suggested that urinary ethanolamine was associated with higher eGFR lending support for a potential causal protective role. This finding, however, should be interpreted with some caution as the association was not evident when using other measurements of kidney function, such as cystatine C based eGFR ($p = 0.016$) or BUN ($p = 0.18$). The association was based on the rs62313082 variant, which was associated with lower *ETNPPL* gene expression in the kidneys. The *ETNPPL* gene encodes for Ethanolamine-Phosphate Phospho-Lyase that catalyses the breakdown of phosphoethanolamine. Ethanolamine is an initial precursor for phosphoethanolamine and for the biosynthesis of two primary phospholipid classes, phosphatidylcholine (PC) and phosphatidylethanolamine (PE), as well as sphingophospholipid and a variety of N-acylethanolamines. The *ETNPPL* gene was recently implicated also in hyperinsulinemia-induced insulin resistance[41].

Furthermore, MR analysis suggested 1-methylnicotinamide as a causal risk factor for BMI. Indeed, serum levels of 1-methylnicotinamide were positively correlated with BMI in observational setting[42], and a caloric restriction and exercise intervention suggested that 1-methylnicotinamide enhances the utilization of energy stores in response to low muscle energy availability[43]. Thus, our findings support the previous suggestion of 1-methylnicotinamide as an early marker for metabolic disease[43]. Altogether, our findings, and the genome-wide metabolite results could be utilized to test and support biological hypotheses originating from observational studies.

Our study included individuals with reduced glomerular filtration rate or albuminuria, potentially enhancing our power to detect some associations, as previous studies have shown that genetic studies on urinary metabolites in individuals with CKD can detect signals that would be harder to detect in the general population alone[6]. It is however important to note, that adding eGFR as a kidney filtration covariate in the GWAS did not have a significant impact on our genetic association results.

As a limitation, the study participants were mostly of European origin and further studies are required to investigate the generalizability of our findings to other populations. The participants from the three included studies also had different clinical profiles with one of the studies including only individuals with T1D; notably, the mean age in the

**Table 3 | Metabolite lead variant associations with eGFR and CKD in the CKDGen meta-analysis[31], and DKD phenotypes in DNCRI-SUMMIT[16] meta-analysis**

| Urinary metabolite | | | eGFR | | CKD | | DKD | | |
|---|---|---|---|---|---|---|---|---|---|
| Chr:Pos:EA:NEA | Rsid | Metabolite | Effect | Effect | P-value | Effect | P-value | Effect | P-value | Phenotype |

| Chr:Pos:EA:NEA | Rsid | Metabolite | Effect | Effect | P-value | Effect | P-value | Effect | P-value | Phenotype |
|---|---|---|---|---|---|---|---|---|---|---|
| 2:211540507:A:C | rs1047891 | Glycine | 0.23 | −0.0065 | $3.6 \times 10^{-64}$ | 0.055 | $2.3 \times 10^{-07}$ | | | |
| 4:109716840:A:T | rs62313082 | Ethanolamine | 0.14 | 0.0019 | $6.1 \times 10^{-08}$ | | | | | |
| 5:1188285:A:G | rs11133665 | Glycine | −0.13 | −0.0016 | $8.6 \times 10^{-05}$ | | | | | |
| 5:1188285:A:G | rs11133665 | Histidine | −0.31 | −0.0016 | $8.6 \times 10^{-05}$ | | | | | |
| 5:1188285:A:G | rs11133665 | Threonine | −0.15 | −0.0016 | $8.6 \times 10^{-05}$ | | | | | |
| 5:1188285:A:G | rs11133665 | Tryptophan | −0.17 | −0.0016 | $8.6 \times 10^{-05}$ | | | | | |
| 5:1188285:A:G | rs11133665 | Tyrosine | −0.3 | −0.0016 | $8.6 \times 10^{-05}$ | | | | | |
| 5:150708711:C:G | rs147000073 | Glycine | −1.1 | −0.0071 | $1.1 \times 10^{-04}$ | 0.12 | $1.2 \times 10^{-02}$ | | | |
| 7:17287998:A:G | rs2106727 | Quinic acid | −0.12 | −0.0022 | $3.5 \times 10^{-10}$ | 0.032 | $9.1 \times 10^{-04}$ | 0.096 | $3.4 \times 10^{-02}$ | ESRD vs. macroalb. |
| 7:17287998:A:G | rs2106727 | Trigonelline | −0.1 | −0.0022 | $3.5 \times 10^{-10}$ | 0.032 | $9.1 \times 10^{-04}$ | 0.096 | $3.4 \times 10^{-02}$ | ESRD vs. macroalb. |
| 9:136146597:T:C | rs550057 | Xylose | 0.2 | 0.0019 | $9.2 \times 10^{-07}$ | | | 0.19 | $1.1 \times 10^{-04}$ | ESRD vs. macroalb. |
| 12:345369:C:G | rs2080403 | 3-Aminoisobutyrate | −0.16 | 0.004 | $8.5 \times 10^{-30}$ | | | 0.058 | $5.1 \times 10^{-03}$ | Any DKD |
| 12:4521511:A:T | rs78470967 | 4-Deoxythreonate | 0.34 | 0.0036 | $1.8 \times 10^{-04}$ | | | | | |
| 12:122344302:A:G | rs1795967 | 2-Hydroxyisobutyrate | −0.51 | 0.0019 | $1.9 \times 10^{-04}$ | | | | | |
| 17:37631883:C:G | rs11078902 | Threonine | −0.12 | 0.0058 | $2.1 \times 10^{-47}$ | −0.041 | $1.3 \times 10^{-04}$ | | | |
| 17:37633970:A:C | rs12453397 | Tryptophan | 0.16 | −0.0057 | $4.9 \times 10^{-46}$ | 0.041 | $1.3 \times 10^{-04}$ | | | |
| 17:37633970:A:C | rs12453397 | Tyrosine | 0.25 | −0.0057 | $4.9 \times 10^{-46}$ | 0.041 | $1.3 \times 10^{-04}$ | | | |
| 17:37636695:T:G | rs4795371 | Histidine | −0.22 | 0.0057 | $1.8 \times 10^{-46}$ | −0.041 | $1.3 \times 10^{-04}$ | | | |

All nominally significant associations ($p < 0.05$) are shown for variants with at least one significant association ($p < 3.5 \times 10^{-04}$, i.e., $p < 0.05$ / 3 phenotypes / 48 SNVs).
*Chr:Pos:EA:NEA* Chromosome position, effect allele, and non-effect allele, *Rsid* variant rs-identifier, *Metabolite* the associated urinary metabolite, *Effect* effect estimate, *P-value* p-value of the association (two-sided test), *Phenotype* the DKD phenotype, *eGFR* estimated glomerular filtration rate, *CKD* chronic kidney disease, *DKD* diabetic kidney disease.

**Table 4 | Two sample Mendelian randomization analysis results with p < 0.05 / 468 = 1.1 × 10⁻⁴ using urinary metabolites as the exposures for outcomes from IEU GWAS database, CKDGen meta-GWAS, DNCRI meta-GWAS, and DIAMANTE meta-GWAS**

| Outcome | Exposure | Rsid | Beta (SE) | P |
|---|---|---|---|---|
| Body mass index (BMI) | 1-Methylnicotinamide | rs17322446 | 0.08 (0.02) | $2.3 \times 10^{-05}$ |
| | 3-hydroxyhippurate | rs6968554 | 0.10 (0.02) | $2.5 \times 10^{-06}$ |
| | 4-Deoxythreonate | rs181558, rs78470967 | 0.05 (0.01) | $7.6 \times 10^{-05}$ |
| | Quinic acid | rs2106727 | 0.08 (0.02) | $2.5 \times 10^{-06}$ |
| | Trigonelline | rs2106727 | 0.10 (0.02) | $2.5 \times 10^{-06}$ |
| eGFR | 3-hydroxyhippurate | rs6968554 | 0.02 (0.00) | $3.7 \times 10^{-10}$ |
| | Ethanolamine | rs62313082 | 0.01 (0.00) | $6.1 \times 10^{-08}$ |
| | Quinic acid | rs2106727 | 0.02 (0.00) | $3.5 \times 10^{-10}$ |
| | Trigonelline | rs2106727 | 0.02 (0.00) | $3.5 \times 10^{-10}$ |
| UACR | 3-hydroxyhippurate | rs6968554 | 0.23 (0.02) | $3.5 \times 10^{-25}$ |
| | Quinic acid | rs2106727 | 0.18 (0.02) | $7.6 \times 10^{-25}$ |
| | Trigonelline | rs2106727 | 0.21 (0.02) | $7.6 \times 10^{-25}$ |

Results involving one variant were obtained using Wald ratio method and two variants with inverse variance weighted method (all p-values derived from two-sided tests). *UACR* urinary albumin to creatinine ratio, *eGFR* estimated glomerular filtration rate.

T1D cohort was 37.7 years vs 49.8 and 55.8 in the two general population cohorts. While the results were mainly concordant across the studies, we found evidence of heterogeneity in the heritability for six metabolites; furthermore, 9 of the 54 COJO lead signals demonstrated significant heterogeneity between the studies, with discrepant effect size directions in the T1D cohort for the glycine *GM2A* association and for 5 secondary 3-aminoisobutyrate signals in the chromosome 5p13.2 locus. We note that some of the metabolites had high missingness in some or all of the cohorts; to the best of our knowledge this missingness is random, however, it affected our power to detect both single variant and Mendelian Randomization associations and may hinder the metabolite applicability as biomarkers at least when using the current methods. Finally, the urinary NMR platform that we utilized was able to detect 54 urine metabolites; while this is significantly more than e.g., the genetic analyses performed for the four urinary laboratory measurements in UK Biobank[8], the number of metabolites that can be captured with the NMR platforms remains far below the width of metabolites detected e.g., with mass spectrometry based methods[7].

Altogether, this work provides a catalogue of genetic associations for 53 metabolites, which can be utilized, for example, to investigate how urinary metabolites are linked to human health and disease risk.

## Methods
### Study cohorts
Generation Scotland (GS) is a family-based population study of 24,000 individuals from across Scotland[11]. Individuals aged between 35 and 65 years were recruited at random from 2006 to 2010 from collaborating medical practices. These participants then identified ≥first degree relatives who would also be able to participate, resulting in a final age range of 18 to 98 years. Participants attended a staffed research clinic where they completed a health questionnaire, had physical and clinical characteristics measured, and fasting blood and urine samples collected according to standard operating procedures. Serum and urine aliquots were stored at -80 °C for future analyses. For this study, we selected a subset of unrelated individuals with GWAS genotype data that passed genotyping quality control conditions and had a spot urine sample available. The selection of samples was random but with an aim to balance the numbers of males and female participants (1373 males, 1370 females), with a total of 2743 individuals in the analysis. All the GS participants gave informed consent and the ethical approval for the study was obtained from the Tayside Committee on Medical Research Ethics (on behalf of the National Health Service, reference 05/S1401/89). The GS study was performed in accordance with the Declaration of Helsinki.

VIKING is a family-based population study of 2108 individuals from the population isolate of the Shetland Isles in northern Scotland[12]. Recruitment ran from 2013 to 2015 with the selection criteria requiring individuals to be ≥18 years and have two or more grandparents born in the Shetland Isles. Over 90% of resulting participants had three or four grandparents from Shetland and most were related individuals from large kindreds. Participants attended clinics where physical characteristics were measured and fasting blood and urine samples were collected according to standard operating procedures. Plasma, serum, whole blood and urine aliquots were stored at -80 °C for future analyses. In this study, we included all samples with GWAS genotype data that passed genotyping quality control conditions and had a spot urine sample available, resulting in total 2024 individuals in the analysis. All the Viking Health Study – Shetland (VIKING) participants gave informed consent and the study was approved by the South East Scotland Research Ethics Committee, NHS Lothian (reference: 12/SS/0151). The VIKING study was performed in accordance with the Declaration of Helsinki.

Finnish diabetic nephropathy study (FinnDiane) is an ongoing (1997->), nation-wide multicentre study focusing on diabetic complications and currently comprises of over 5000 adults with type 1 diabetes[9,10]. In this study we included individuals with genotype data and urinary metabolite data measured from 24 h urine collection, except one overnight urine collection, stored at -20C°. In addition, to assure correct diagnosis of T1D, we required age at onset of diabetes <40 years and insulin treatment initiated within one year from diabetes diagnosis. Moreover, since kidney function may affect the urinary metabolite levels, we excluded Individuals with prevalent end-stage kidney disease (ESKD), defined as kidney transplantation or dialysis treatment, and individuals with eGFR <10 mL/min/1.73m², at urine collection day. Finally, 3244 individuals were included in the analysis. The FinnDiane study protocol was approved by the Ethical Committee of the Helsinki and Uusimaa Hospital District (491/E5/2006, 238/13/03/00/2015, and HUS-3313-2018, July 3rd, 2019) and the participants gave their informed consent before recruitment. The FinnDiane study was performed following the Declaration of Helsinki.

All three cohorts had a balanced distribution of men and women (Supplementary Table 1).

### Metabolite quantification by NMR
Urinary metabolomics by NMR was measured for all available and qualifying FinnDiane and VIKING study participants, and for 2743 of

the ~20,000 Generation Scotland participants with urine sample available (Supplementary Fig. 10). The urinary metabolite quantification has been described previously[2]. Briefly, metabolite quantification of the urine samples was performed using a proprietary NMR metabolite profiling service (Nightingale Health, Helsinki, Finland). The NMR-based measurements were conducted from 500 µl of stored samples using a 600 MHz Bruker AVANCE III HD NMR spectrometer (Bruker BioSpin, Switzerland) with automated sample changer and cryoprobe. The spectral data were acquired using standard water-suppressed acquisition settings. The sample preparation and NMR acquisition parameters were designed for high-throughput initially selecting metabolites based on feasibility for automated quantification. This approach emphasises metabolites at high abundance in urine, and those which generate minimal signal overlap in the proton NMR spectrum. As such, the metabolite selection was not based on prior biological relevance of the selected metabolites or emphasis of certain metabolic pathways. In each study, the metabolite concentrations below the detection limit were set to the observed minimum for each metabolite and missing values were left missing. The urinary metabolite concentrations were divided by urinary creatinine concentration to normalise for urine volume.

## Genotyping and imputation

GS and VIKING samples were genotyped using the Illumina HumanOmniExpressExome-8v1-2 chip (Illumina, San Diego, CA) and individuals with a call rate of ≤ 98% and SNVs with a call rate of ≤ 98%, Hardy–Weinberg equilibrium $p \leq 1 \times 10^{-06}$ and a minor allele frequency (MAF) of ≤ 1% were excluded during quality control. Phasing was carried out using SHAPEIT (v2 r837) and imputation was performed using the Haplotype Reference Consortium reference panel (HRC.r1-1) on the Sanger Imputation Server with the PBWT software. Post imputation quality control excluded duplicate and monomorphic variants and SNVs with an imputation quality score of <0.4.

FinnDiane samples were genotyped using the HumanCoreExome-12 v1.0, -12 v1.1, and -24 v1.0 BeadChips (Illumina, San Diego, CA). The quality control and data processing has been described in more detail before[44,45]. In short, SNVs with call rate of ≤ 95% or excessive deviation from Hardy-Weinberg equilibrium were excluded, Haplotypes were phased with SHAPEIT (v2 r837) and genotypes imputed with Minimac3 (v1.0.14) using 1000 Genomes phase 3 version 5 as the reference panel.

## Heritability analysis

The heritability of the urinary metabolites was analysed in each cohort. In FinnDiane, the GCTA tool (v1.93.2beta) was utilized to estimate the genetic relationship matrix which was filtered to not include any individuals with relatedness greater than 0.025. The variance explained by all the SNVs was estimated by restricted maximum likelihood (REML) analysis (GCTA-GREML) using the default options and adjusting for age, sex, eGFR, genotyping batch, and two first genetic principal components[46,47]. In GS and VIKING heritability was estimated using a variance component model available within the RegScan GWAS pipeline. The heritability estimates were meta-analysed with random-effects model utilizing the inverse variance method and between study variance $\tau^2$ was estimated with restricted maximum-likelihood estimator. The between study heterogeneity was tested with Q-test. Analysis was performed with R-package meta (v.6.5-0).

## GWAS analysis, meta-analysis with Metal, and GCTA-COJO

Study-level GWAS analysis was conducted separately for each cohort and the results were first quality controlled and harmonized before meta-analysis, and finally, a conditional joint analysis was performed to identify SNVs independently associated with urinary metabolites. Before GWAS analysis urinary metabolite to creatinine ratios and creatinine values were regressed on the covariates and the residuals were inverse normal transformed.

GS and VIKING GWAS were performed using RegScan (v 0.5) accounting for relatedness within each cohort[48]. The analysis model included age and investigator-assigned sex as covariates. FinnDiane GWAS was executed with SNPTest (v2.5.2). Before the analysis, first-degree relatives were removed preferring individuals with most complete metabolite data until no first-degree relative pairs were left in the data set. The analysis model included age, investigator-assigned sex, genotyping batch and two first genetic principal components as covariates. The association of genetic variants with urinary metabolites was tested with score test with an additive model based on genotype probabilities to account for genotype uncertainty. All p-values were derived from two-sided tests.

Before the meta-analysis study-level quality control was performed with EasyQC R-package (v9.2, www.genepi-regensburg.de/easyqc)[49]. First, any association results with missing or implausible data, monomorphic variants, and variants with imputation quality <0.4 were removed, second, allele coding and marker names were harmonized and possible duplicates were removed, finally, variants were checked against the appropriate reference data and any variants with mismatching alleles or allele frequency difference >0.2 compared to the reference were removed.

Meta-analysis of the individual GWAS was performed using METAL software (version 2011-03-25) applying inverse variance weighted method and genomic control correction for the individual study level results[50].The results were filtered to include variants with MAF ≥ 0.01 and found at least in 2 out of 3 studies. Genomic inflation factors $\lambda_{GC}$ were calculated for the meta-analysis results (Supplementary Data 13). Signals for the same metabolite were considered distinct if they were at least 3Mbp apart.

Approximate conditional and joint GWAS analysis was performed to identify SNVs independently associated with urinary metabolites applying the GCTA-COJO software (v1.93.2beta)[13,46]. The filtered METAL results were used as the input and the whole FinnDiane cohort ($n = 6,019$) was used as the reference population to estimate LD. Default options were used to perform stepwise model selection to select independently associated SNVs. Association results for glucose were spurious and are not reported.

The regional association signal around the COJO lead variants was visualized using LocusZoom stand-alone software (v1.4, http://genome.sph.umich.edu/wiki/LocusZoom_Standalone). The LD information was calculated using the 1000 Genomes phase 3 European population.

The effect of different covariate adjustment strategies were tested in FinnDiane for the COJO lead signals by analysing two additional models: 1) eGFR was added as a covariate before the INT of the residuals prior to the genetic association analysis (Supplementary Fig. 9A), and 2) age and sex were included as covariates in the genetic association analysis to test if age and sex are affecting the observed metabolite associations by re-introduced covariate effects after the INT of the residuals (Supplementary Fig. 9B)[51].

## Annotation of the COJO lead variants

The lead SNVs from COJO analysis were annotated with genes and variant effects with Ensembl Variant Effect Predictor (VEP) web tool (Assembly GRCh38.14 version 110). The SNVs were queried for all consequences. The most severe consequence per SNV and gene was selected based on the severity as estimated by Ensembl (www.ensembl.org/info/genome/variation/prediction/predicted_data.html accessed 2023-11-14). If the variant was not located in a transcript the closest gene was selected. In addition, likely relevant target genes were queried form the literature and from the EGEA database (https://github.com/fauman/EGEA) for genes within 500 kbp from the lead variant.

## Replication

We tested replication of the 33 novel signals for 16 metabolites based on two previous urinary metabolite studies, with 11 out of the 16 metabolites found in a mass spectrometry-based study with 1,399 metabolites measured in 4,912 individuals[7] and 9 out of the 16 metabolites found in an NMR-based study with 55 targeted metabolites measured in 5,552 individuals[4]. We found data for 24 of the 33 signals in either of the two studies. Summary statistics for Schlosser et al. 2023 were downloaded from the NHGRI-EBI GWAS Catalog[52] on 2024-04-24 and for Raffler et al. 2015 from http://metabolomics.org/gwas on 2024-05-13.

## Additional phenotypic data

Kidney function was quantified by eGFR calculated with the CKD-EPI formula[53] from serum and plasma creatinine values, and, in addition, by urinary albumin excretion rate (AER) in the FinnDiane cohort: normal AER (AER ≤ 30 mg per 24 h), moderate albuminuria (30 mg per 24 h <AER ≤ 300 mg per 24 h), and severe albuminuria (AER > 300 mg per 24 h). Albuminuria category was determined as the highest category in at least 2 out of 3 consecutive determinations.

## eQTL analysis in kidney and whole blood

Expression quantitative trait locus (eQTL) associations in cis were queried for the variants associated with urinary metabolites. We utilized kidney eQTL data from microdissected human kidney tubule ($n = 356$) and glomeruli ($n = 303$) samples[54], and meta-analysis of 686 kidney samples[22] downloaded from https://susztaklab.com/Kidney_eQTL/download.php. Cis-eQTL associations in whole blood were queried from eQTLGen data set[24] (https://eqtlgen.org/cis-eqtls.html and IEU GWAS database).

In the kidney data sets we identified eQTLs and their target genes at the urinary metabolite lead variants and any additional eQTLs at proxies of the lead variants ($r^2 > 0.8$) using R-package LDlinkR (v.1.2.3). In total, we found 94 candidate eQTL target genes and selected eQTLs with $p < 0.05 / 94 = 5.26 \times 10^{-4}$. However, we only had access to kidney eQTL data sets pre-filtered to include signals with false discovery rate (FDR)-adjusted $p < 0.05$ (tubule and glomeruli) and FDR-adjusted $p < 0.01$ (kidney meta-analysis) and consequently all candidate eQTLs were significant.

In the whole blood cis-eQTL data we identified 931 eQTLs at the lead variants and selected eQTLs with $p < 0.05 / 931 = 5.4 \times 10^{-5}$ as significant. Furthermore, we tested for colocalization of cis-eQTL signal for gene expression in blood with urinary metabolite signals. First, we identified genes with cis-eQTLs at the lead loci, if no genes were found we selected all genes with cis-eQTLs within 100kbp from the lead locus. Second, we tested colocalization between the genes eQTL signal and the urinary metabolite signal in a region extending 250kbp from the lead locus with R-package coloc (v.5.1.0.1). More specifically, we used the Bayesian colocalization analysis assuming one causal variant for each trait implemented in coloc.abf function, and calculated a posterior probability (PP) for one common causal variant[55]. Signals were considered colocalized if PP > 0.5.

## GWAS look-ups: GWAS catalog, CKDGen, and DNCRI-SUMMIT

Lead variant associations were queried (2023-09-21) for previously reported associations from the GWAS catalog using R packages LDlinkR (v.1.2.3) and gwasrapidd (v.0.99.14). We included previous associations with $r^2 > 0.8$ within -/+500,000 bp from the lead variant and with $p < 5 \times 10^{-8}$ using the 1000 Genomes European population as the reference. The previously reported traits were classified as urinary metabolite, blood metabolite or another trait by searching for key words in the phenotype description and p-value annotation. Urinary metabolite traits were matched with regular expression: 'urinary metabolite', and blood metabolites with 'serum metabolite|blood metabolite|serum uric acid levels|blood urea nitrogen levels'. Furthermore, targeted lookups were performed for kidney-related traits from the CKDGen consortium meta-analyses on CKD and eGFR in individuals with European ancestry from the general population[31] available from https://ckdgen.imbi.uni-freiburg.de; and for DKD phenotypes from the DNCRI-SUMMIT meta-analysis[16] available from https://t2d.hugeamp.org/downloads.html.

## FUMA

FUMA v.1.3.7[56] web interface was used to perform MAGMA v1.08 tissue specificity (GTEx v8) and gene set enrichment analysis. SNVs were mapped to the protein coding genes within 10 kb windows (with unique Ensembl ID). For the gene set enrichment analysis, 15,496 gene sets (5500 curated gene sets (9 data resources including KEGG, Reactome and BioCarta), 9996 GO terms (biological processes (bp), cellular components (cc) and molecular functions (mf))) from MsigDB v7.0 were included and run with default parameters. Gene-sets with p-value $< 3 \times 10^{-6}$ were defined as significant by MAGMA Bonferroni correction. For tissue expression analysis, gene expression data sets were obtained from GTEx v8. MAGMA gene-property test was performed for average gene-expression per category (e.g. tissue type) conditioning on average expression across all categories (one-side) to test the positive relationship between gene expression in a specific tissue and genetic associations.

In addition, gene set enrichment was performed using the GEN-E2FUNC tool in FUMA v.1.3.7[56]. Genes annotated to SNVs with a $p < 1 \times 10^{-5}$ were used as input. The list of genes was compared to a set of 19,283 background genes using hypergeometric tests to determine the overrepresentation of biological functions. A minimum number of two genes per gene set and an FDR Benjamini-Hochberg adjusted $p < 0.05$ were required for gene sets to be reported.

## Mendelian randomization analysis

We performed two-sample Mendelian randomization analysis to test if kidney function, measured by estimated glomerular filtration rate (eGFR) or urinary albumin creatine ratio (UACR), causally affects urinary metabolite concentrations, and conversely, if urinary metabolite levels causally affect kidney function or other traits including type 2 diabetes (T2D), BMI, and kidney disease (Supplementary Table 4). We additionally tested whether BMI causally affected the urinary metabolites.

To ensure that the instruments are strongly associated with each respective exposure (relevance assumption), we selected only genetic variants as instruments that showed a strong and independent association with the exposure. Therefore, we included only variants that were genome-widely and independently associated with the exposure ($r^2 = 0.001$, $p < 5 \times 10^{-8}$). Furthermore, for each genetic variant we estimated the F-value using the formula beta$^2$ per (standard error)$^2$[57] and included only IVs with a $F > 10$.

In the first analysis we used two kidney function markers, eGFR and UACR, as the exposures and urinary metabolites as the outcomes. As the instrumental variables (IV) for eGFR we used 225 variants associated with eGFR (150 independent variants after clumping) in a European ancestry sub-analysis with 567,460 individuals ($p < 5 \times 10^{-8}$) from the CKDGen consortium[31], and as the IV for UACR we used 61 variants associated with UACR (51 independent variants after clumping) in the European ancestry analysis with 547,361 individuals ($p < 5 \times 10^{-8}$) from the CKDGen consortium[58]. For BMI, we used 458 variants associated with BMI (439 independent variants after clumping, 419 after effect allele harmonizing) in UK biobank available in the manually curated OpenGWAS database[59] (Dataset: ukb-b-19953).

In the second analysis, we employed urinary metabolites as the exposures. We selected IVs to be the variants associated with urinary metabolites in the COJO analysis with $p < 5 \times 10^{-8}$ (Supplementary Data 14). As outcomes we used 10 DKD traits from the JDRF DNCRI GWAS[44] (downloaded from https://t2d.hugeamp.org/downloads.

html), 3 traits from CKDGen consortium related to kidney function (downloaded from https://ckdgen.imbi.uni-freiburg.de), T2D from the DIAMANTE consortium (https://t2d.hugeamp.org/downloads.html), and 6 traits from the IEU GWAS database (https://gwas.mrcieu.ac.uk/, Supplementary Table 4).

Both analyses were performed with TwoSampleMR R package (v.0.6.4). Shortly, the genetic variants for the exposures were first clumped using a 10,000 kb window, a clumping R-square cut-off of 0.001, and 1000 Genomes European samples to estimate LD. Second, the effect alleles were harmonized between the exposure and outcome GWASes, and finally MR analysis was performed with default MR methods in the TwoSampleMR package: We used inverse variance-weighted (IVW) regression if at least 2 variants remained as valid IVs for the exposure, or Wald's ratio test if only one variant was available. For exposures with 3 or more IVs, causality was further assessed using methods less sensitive to pleiotropy. These methods assume that only a subset of the variants are valid instruments (i.e., not pleiotropic), including weighted median, simple and weighted mode MR methods. We also used MR-Egger, which allows all variants to have pleiotropic effects, under the InSIDE (Instrument Strength Independent of Direct Effect) assumption. We additionally tested if the Egger intercept deviated from zero, using the Egger intercept test, as this may indicate either horizontal pleiotropy or a violation of the InSIDE assumption underlying MR-Egger[60]. For MR associations with multiple IVs, we used the Cochran Q test to examine any heterogeneity of SNVs' estimates. All the steps were performed using the default options.

As the commonly used serum creatinine based eGFR can also reflect the creatinine metabolism in addition to kidney function, we performed sensitivity analyses for the Mendelian Randomization using two other available markers of kidney function, namely cystatine C based eGFR (eGFRcys; GWAS in 480k European ancestry individuals[21]), and blood urea nitrogen (BUN; GWAS in 240k European ancestry individuals[31]). For the first analysis using kidney function as exposure, and the urinary metabolites as the outcomes, we adopted an approach similar to Wuttke et al. [31] by including as IVs only the eGFR-associated genetic variants ($p < 5 \times 10^{-8}$) with additionally having directionally consistent and nominally significant p-value ($p < 0.05$) with i) eGFRcys or ii) BUN. For the second analysis, using urine metabolites as exposures, and kidney function as outcome, we alternatively used i) eGFRcys or ii) BUN as the outcome measure.

### Reporting summary

Further information on research design is available in the Nature Portfolio Reporting Summary linked to this article.

### Data availability

The GWAS meta-analysis summary statistics for the 54 studied metabolites except glucose have been deposited in the GWAS Catalog under accessions codes GCST90451702–GCST90451754 (https://ebi.ac.uk/gwas). For VIKING, the research data used in this study are available through managed access by application (accessQTL@ed.ac.uk), following approval by the QTL Data Access Committee with expected timeframe for response of about 2 months (https://viking.ed.ac.uk/). These data are available under managed access due to the consent given by the participants and Research Ethics Committee approvals. Each approved project is subject to a data or materials transfer agreement (D/MTA) or commercial contract. Data may be shared with academic or commercial recipients worldwide and may be used within the parameters of the Research Ethics Committee approvals. The individual-level GWAS and metabolite data for the Generation Scotland study are available under managed access for legal restrictions, access can be obtained by a proposal to the access committees of Generation Scotland; access@generationscotland.org; Each approved project is subject to a data or materials transfer agreement (D/MTA) or commercial contract. The raw individual-level data of the FinnDiane study are protected and are not available due to data privacy laws and participants written consent, which does not allow sharing of such data. The processed summary statistics for FinnDiane study can be obtained by a proposal to the corresponding author. The GWAS summary statistics for general population kidney traits from the CKDGen consortium used in this study are available on the CKDGen project website (https://ckdgen.imbi.uni-freiburg.de); DKD phenotypes and T2D available in the T2D knowledge portal under "GENIE 2022 DKD GWAS" and "DIAMANTE T2D 2022 GWAS" (https://t2d.hugeamp.org/downloads.html); BMI available in the OpenGWAS database under accession code ukb-b-19953 (https://gwas.mrcieu.ac.uk/datasets/); asthma and coronary heart disease in the IEU GWAS database under finn-b-J10_ASTHMA and ieu-a-7, respectively (https://gwas.mrcieu.ac.uk/). The kidney eQTL data used in this study are available in the Human Kidney eQTL Atlas (https://susztaklab.com/Kidney_eQTL/download.php); and whole blood cis-eQTL data available on the eQTLGen website (https://eqtlgen.org/cis-eqtls.html).

### Code availability

No custom code or algorithm was developed for the analyses performed in this study, instead previously designed, publicly available, pipelines and code were used (as described in the Methods): SNPTest (https://www.chg.ox.ac.uk/~gav/snptest/), EasyQC (https://www.uni-regensburg.de/medizin/epidemiologie-praeventivmedizin/genetische-epidemiologie/software/index.html), RegScan (https://bioinformaticshome.com/tools/gwas/descriptions/RegScan.html#gsc.tab=0), METAL, GCTA-COJO (https://yanglab.westlake.edu.cn/software/gcta/#COJO), Locuszoom (https://genome.sph.umich.edu/wiki/LocusZoom_Standalone), GCTA GREML https://yanglab.westlake.edu.cn/software/gcta/#GREML, meta (https://CRAN.R-project.org/package=meta), LDlinkR (https://cran.r-project.org/web/packages/LDlinkR/vignettes/LDlinkR.html) gwasrapidd (https://cran.r-project.org/web/packages/gwasrapidd/vignettes/gwasrapidd.html), coloc (https://chr1swallace.github.io/coloc/articles/vignette.html), FUMA https://fuma.ctglab.nl/, TwoSampleMR (https://mrcieu.github.io/TwoSampleMR/).

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

## Acknowledgements

The research in FinnDiane study was supported by funding from Folkhälsan Research Foundation (PHG), Wilhelm and Else Stockmann Foundation (PHG), Liv och Hälsa Society (PHG), Helsinki University Hospital Research Funds (EVO TYH2018207; PHG), Academy of Finland (299200 (NS), and 316664 (PHG)), Novo Nordisk Foundation (NNF OC0013659 (PHG), NNF23OC0082732 (NS)), Sigrid Jusélius Foundation (PHG and NS), and Finnish Diabetes Research Foundation (PHG). Genotyping of the FinnDiane GWAS data was funded by the Juvenile Diabetes Research Foundation (JDRF) within the Diabetic Nephropathy Collaborative Research Initiative (DNCRI; Grant 17-2013-7, co-investigator PHG), with GWAS quality control and imputation performed at University of Virginia. Generation Scotland received core support from the Chief Scientist Office of the Scottish Government Health Directorates [CZD/16/6, co-investigator DJP] and the Scottish Funding Council [HR03006 for DJP] and is currently supported by the Wellcome Trust [216767/Z/19/Z for DJP]. Genotyping of the GS:SFHS samples was carried out by the Genetics Core Laboratory at the Edinburgh Clinical Research Facility, University of Edinburgh, Scotland and was funded by the Medical Research Council UK (MC_PC_U127561128 for CH&JFW) and the Wellcome Trust (Wellcome Trust Strategic Award "STratifying Resilience and Depression Longitudinally" (STRADL) Reference 104036/Z/14/Z). CH was supported by the MRC Human Genetics Unit quinquennial programme grant "QTL in Health and Disease" (MC_UU_00007/10 for CH). The Viking Health Study – Shetland (VIKING) was supported by the MRC Human Genetics Unit quinquennial programme grant "QTL in Health and Disease" (MC_UU_00007/10 for CH). We acknowledge the skilled technical assistance of Heli Krigsman, Hanna Olanne, Maikki Parkkonen, Mira Korolainen, Anna Sandelin, Jaana Tuomikangas, and Kirsi Uljala (Folkhälsan Research Center, Finland), and all the physicians and nurses at each FinnDiane study center taking part in the enrolment and clinical characterization of the participants (Supplementary Table 5 for a list of study centers and investigators involved in the FinnDiane study). We are grateful to all the families who took part, the general practitioners and the Scottish School of Primary Care for their help in recruiting them, and the whole Generation Scotland team, which includes interviewers, computer and laboratory technicians, clerical workers, research scientists, volunteers, managers, receptionists, healthcare assistants and nurses. The Viking Health Study DNA extractions and genotyping were performed at the Edinburgh Clinical Research Facility, University of Edinburgh. We would like to acknowledge the invaluable contributions of the research nurses in Shetland, the administrative team in Edinburgh and the people of Shetland. For the purpose of open access, the author has applied a Creative Commons Attribution (CC BY) licence to any Author Accepted Manuscript version arising from this submission.

## Author contributions

EV contributed to design of the study, data analysis, interpretation of the results, and drafted the manuscript. AR and EHD contributed to data analysis, manuscript writing, and interpretation of the results. SM contributed to metabolite data quantification, manuscript writing and interpretation of the results. DJP and AC contributed to data acquisition. JFW contributed to data acquisition. PHG contributed to design of the study. CH and NS contributed to design of the study, data analysis, interpretation of the results, and manuscript writing. All authors critically read and approved the final version to be submitted and published, and agree to be accountable for all aspects of the work in ensuring that questions related to the accuracy or integrity of any part of the work are appropriately investigated and resolved.

## Competing interests

P-H G has served on advisory boards for AbbVie, Astellas, AstraZeneca, Bayer, Boehringer Ingelheim, Cebix, Eli Lilly, Janssen, Medscape, MSD, Mundipharma, Nestlé, Novartis, Novo Nordisk, Sanofi, and has received lecture honoraria from Astellas, AstraZeneca, Bayer, Berlin Chemie, Boehringer Ingelheim, Eli Lilly, Elo Water, Genzyme, Medscape, Menarini, MSD, Mundipharma, Novartis, Novo Nordisk PeerVoice Sanofi, and Sciarc. P-H G has also received investigator-initiated grants from Eli Lilly and Roche. S M received lecture honoraria from Encore Medical Education. All other authors declare no competing interests.

## Additional information

## FinnDiane Study Group

Erkka Valo [1,2,3], Stefan Mutter[1,2,3], Emma H. Dahlström [1,2,3], Per-Henrik Groop [1,2,3,7,8] & Niina Sandholm [1,2,3,9] ✉

A full list of members and their affiliations appears in the Supplementary Information.

