## [Transparent Peer Review file · Nature Communications]

Genome-wide characterization of 54 urinary metabolites reveals molecular impact of kidney function

Corresponding Author: Dr Niina Sandholm

Version 0:

Reviewer comments:

Reviewer #1

(Remarks to the Author)

Erkka and colleagues present a nicely written and extensive manuscript on the genetics of 54 urine metabolites, illuminating casual relations between metabolites and kidney function. They identified novel loci and intriguing causal links. While I enjoyed reading the paper, I have to highlight several methodological considerations, which I hope the Authors can address.

Main issues

1. The main outcome of each GWAS were the urinary metabolite to creatinine ratios. GWAS of ratios are driven by effects on the denominator (creatinine in this case) and I am afraid that many results might reflect creatinine metabolism: <https://www.biorxiv.org/content/10.1101/2023.10.27.564385v1.full>
In fact, many identified variants are associated with higher UACR and higher eGFR_{crea}. UACR and eGFR_{crea} can be high when there is less circulating serum creatinine (causing higher eGFR_{crea}) and, consequently, lower creatinine concentration in the urine (causing higher UACR values for fixed albumin concentrations). See also point 15 below.
2. After regressing the ratios on covariates, GWAS were conducted on the inverse normal transformation of the residuals. It is unclear if adjustment was also applied a second time, to prevent reintroduction of covariate effects (<https://www.nature.com/articles/s41431-018-0159-6>)
3. In the GWAS methods, it is described that "The association of genetic variants with urinary metabolites was tested using a frequentist test ..." Please, define the frequency test.
4. It is unclear whether the genomic control correction was applied twice (at the individual study level and after the meta-analysis) or only before the meta-analysis. If meta-analysis results were not corrected, the genomic inflation factor of each meta-analysis should be reported in the supplement.
5. This study is based on meta-analyses of GWAS from very different studies: the FinnDiane on T1D patients and two general population studies from Scotland. The three studies have different phenotypic and covariate distributions, different genotype platform, etc. For these reasons, between-study heterogeneity statistics should be estimated, reported, and discussed in the case they affect any of the main findings and/or variants used as instrumental variables in Mendelian randomization (MR) analyses.
6. If replication studies are available, the Authors may consider testing their novel findings for replication in independent studies (e.g.: Schlosser 2023, if applicable).
7. As displayed in Suppl. Tab. 2, several metabolites are affected by a high proportion of missing values. Please, clarify if values are missing at random or if missingness is informative (eg: concentrations lower than the limits of detection). In the latter case, this would affect the distribution of the metabolites, reducing range and variability and data would be better imputed. This might also affect MR analyses involving the traits with largest share of missing data.
8. There is an emphasis on kidney function. Please, note that Wuttke et al. 2019 validated their findings on eGFR_{crea} against BUN, to identify a subset of loci that were more likely to pertain to kidney function rather than creatinine metabolism. Data on eGFR_{cys} are also available from the CKDGen Consortium. Those data may be relevant to clarify if the observed causal pathways belong to the kidney function or relate to creatinine metabolism (e.g.: discussion sentence "On the other hand, MR analysis suggested that urinary ethanolamine was associated with higher eGFR lending support for a potential causal protective role").
9. I might have missed the point, but I don't see a high value in presenting extensive heritability analyses, especially the between-cohort comparison, where heterogeneity is also provided. Genetic heritability is heterogenous across different contexts. In addition, with just 3 studies heterogeneity is typically underestimated and estimates not reliable.

10. eQTL analysis covers an extensive part of the Results and becomes difficult to follow. To identify the most likely causal gene underlying detected signals, I would suggest trying to present eQTL analysis and colocalization with gene expression analyses in a more organized way, setting some rules of thumb based on clear cutoffs to identify the most likely causal gene.
11. It is not clear what the purpose of this analysis was: “As eGFR causally affected multiple metabolites we tested if adjusting the GWAS analysis with eGFR affected the metabolite associations in FinnDiane. However, adjustment with eGFR had little effect on the effect estimate for the 41 COJO lead variants included in the FinnDiane GWAS data set (Suppl. Tab. 7).”
12. For MR analyses, instrumental variables should be selected if they have an F statistic ≥ 10 .
13. To support the MR analyses, evidence that the selected instrumental variables satisfy the key assumptions is needed. This is especially important for the IVs used for the metabolites, as they may be influenced by lifestyle and thus violate the assumption of no hidden confounding between IV and exposure. How the Authors excluded the presence of horizontal pleiotropy is also important information.
14. For use of the MR-Egger, are the INSIDE assumptions met? <https://journals.plos.org/plosgenetics/article?id=10.1371/journal.pgen.1010166>
15. The Authors observed that higher urinary 3-hydroxyhippurate, quinic acid, and trigonelline concentrations are causally associated with higher BMI, lower urinary creatinine concentration, higher UACR and higher eGFR estimated from serum creatinine. Given higher levels of the 3 metabolites cause lower urinary creatinine, if urinary albumin is not altered, this would cause higher UACR. And if the 3 metabolites lower urinary creatinine through lowering serum creatinine, that would be compatible also with higher eGFR_{crea}. All of this does not necessarily implicate kidney health.
16. About the causal effect on BMI, checking for reverse causation would be good, as BMI might play the role of a confounder (see an example in PMID 31306056). On the other hand, given the causal effect of BMI on eGFR (direct) and UACR (mediated by DM) (PMID 36305100), multivariate MR of the 3 metabolites on kidney health through BMI would be informative.
17. Finally, the Authors note that all 3 metabolites are found in coffee. Coffee consumption is related to human behavior and is typically related to other social factors such as education, alcohol consumption, smoking. In this case, it is recommended to exclude violation of MR assumptions for all IVs related to the three metabolites.

Minor issues:

18. Introduction: compared to e.g. Schlosser et al 2023, I wouldn't say that 54 metabolites correspond to an “extensive molecular coverage”
19. Introduction: it is not a prerequisite of MR that the exposure and the outcome should be associated to test for a causal effect. Sometimes, the association cannot be seen for the presence of confounders. Maybe, list the three basic hypotheses of MR, which should hold for valid inference.
20. Introduction: the 4th paragraph reads more like a discussion than an introduction.
21. Results: what to the Authors mean with “study-wide”?
22. Results: the citrate-associated rs11567842 was tested for association with several traits. I think that a P-value of 0.012 with eGFR_{crea} from a meta-analysis of >700,000 individuals is not really meaningful.
23. I am not sure I fully agree with sentence: “However, all significant MR analysis findings were based on only one or two significant variants available for each metabolite and need to be interpreted with caution; the largest number - 8 genetic variants – were available for MR for 3-hydroxyisovalerate, which was not associated with any of the studied outcomes ($p > 0.01$)”. Power to detect a causal association also depends on the strength of the IV and their specificity. For molecular data we are often in the presence of a single variant with large effect on the exposure.
24. Discussion: I didn't understand the sentence: “This finding suggests that the glomerular filtration rate needs to be considered when investigating these metabolites as potential biomarkers of disease risk”. Can the Authors try to rephrase?
25. Study limitation section can be expanded.
26. Methods: pleiotropy and heterogeneity are two different things; there might be situations where heterogeneity does not implicate pleiotropy.

Reviewer #2

(Remarks to the Author)

The present study aimed to identify genetic influences on the concentrations of 54 urinary metabolites and evaluate their impact on renal function using genome-wide association studies (GWAS) across three European cohorts, meta-analyses, and Mendelian randomization analysis. Specifically, the authors identified 26 chromosomal regions associated with at least one of the 54 metabolites analyzed. They identified 52 associations in 19 of the 54 studied metabolites, of which 31 were novel. These variants, primarily located in regulatory regions, show a strong association with gene expression in renal tissue, tubules, and glomeruli. Furthermore, subsequent Mendelian randomization analysis suggests that the estimated glomerular filtration rate (eGFR) causally influences 13 urinary metabolites, and there is an association between urinary ethanolamine and high eGFR levels, suggesting a potential protective role. This association is based on rs62313082, which correlates with higher urinary ethanolamine concentrations, higher eGFR, and lower expression of the ETNPPL gene in the kidneys.

This study provides a starting point for future analyses to support experimental validation or biological hypotheses originating from observational studies.

Overall, the article presents a highly complex and well-articulated analysis in all its parts, with interesting and well-exposed results; however, clarifying the workflow of the entire work would be beneficial.

In the results section, the total number of individuals analyzed (8026) does not match the sum of the numbers of individuals in the three cohorts reported in the first paragraph of the results and in Supplementary Table 1. Additionally, in Figure 1, the number of individuals is incorrect (the number of individuals reported in the VIKING cohort is 2077 instead of 2027 compared to the previous data).

In the Methods section, I appreciate the detailed description of the cohorts examined. However, for a more accurate evaluation, it would be advantageous to clearly understand the exclusion criteria applied to each cohort. Therefore, specifying the exclusion criteria used, including the actual number of excluded individuals rather than providing approximations, would be beneficial.

From Supplementary Table 2, it is observed that some metabolites have a high number of missing readings, with different percentages between the two cohorts (e.g., mannitol and taurine). Has this kind of issue been addressed in any way?

Two notes: In line 47, the number of previously unreported associations was 31, not 32. In lines 623, 642, and 645 the GCTA tool is referred to as the GTCA .

Version 1:

Reviewer comments:

Reviewer #1

(Remarks to the Author)

I would like to thank the Authors for considering all points raised, recognizing that they implicated substantial amount of work. I still one major comment (the first one) and a few minor issues (the remainder) to highlight:

1. I appreciate the response to my previous point #2, but I think the evidence that the residuals of 17 out of 54 metabolites were still associated with age and sex confirms the alert by Pain et al (<https://www.nature.com/articles/s41431-018-0159-6>) that normalization may reintroduce the covariate effect. I agree that models should not be adjusted for modifiable factors, but it would be important to demonstrate that age and sex had no role in determining the observed associations, especially given downstream causal analyses involved kidney function, which is related to age and sex.

2. Regarding my previous point #6, significant association in the opposite direction does not constitute replication. Replication should be evaluated with one-sided tests. Maybe it should be commented why association of the same variant with the same metabolite can be significant in the opposite direction.

3. Regarding my previous point #7, I agree with the solution chosen by the Authors to impute missing data to the limit of detection but please, realize that this contradicts the assumption of "missingness at random": in fact, missingness is informative and missing data can reliably be assumed to represent low concentrations. I recommend revising the comment in the manuscript.

4. Great that the Authors analyzed urinary creatinine separately. Suppl. Fig. 4 shows that for most metabolites, the association with creatinine was nearly null, supporting that for most of them the signals are driven by the metabolite levels. However, despite not being significant after multiple-testing correction, for a couple of them the effect on creatinine is not negligible (eg the lowest blue point, maybe 3-Hydroxyisovalerate? Difficult to recognize by the legend). Consequently, I would suggest using a less definitive wording in the comment, by saying eg that supporting that the observed signals are *generally/mostly* driven by the urinary metabolite levels (or similar).

5. The claimed number of metabolites is 54 but conclusions (paper and abstract) refer to 53, please clarify

6. Suppl. Tab. 6: what does the yellow highlight represent?

7. Suppl. Tab. 11: what does boldface of last column items represent?

8. Suppl. Tab. 12: please, fix a typo in the title ("significant")

Reviewer #2

(Remarks to the Author)

The authors responded properly and completely to the reviewers' comments. However, in the manuscript, in lines 707, 710 and 751, the name of the GTCA tool is incorrect. Please correct it to GCTA.

Version 2:

Reviewer comments:

Reviewer #1

(Remarks to the Author)

I have no further concerns. I'd like to thank the Authors for addressing my last points. Great to see that age and sex did not affect the results. Regarding point 2, the replication criterion is now explained clearly and no further action is needed (please, note that most replications are done through lookups and once a two-sided p-value and direction are available, as in this case, deriving the one-sided p-value is straightforward).

REVIEWER COMMENTS

Reviewer #1 (Remarks to the Author):

Erkka and colleagues present a nicely written and extensive manuscript on the genetics of 54 urine metabolites, illuminating casual relations between metabolites and kidney function. They identified novel loci and intriguing causal links. While I enjoyed reading the paper, I have to highlight several methodological considerations, which I hope the Authors can address.

Main issues

1. The main outcome of each GWAS were the urinary metabolite to creatinine ratios. GWAS of ratios are driven by effects on the denominator (creatinine in this case) and I am afraid that many results might reflect creatinine metabolism: <https://www.biorxiv.org/content/10.1101/2023.10.27.564385v1.full>

In fact, many identified variants are associated with higher UACR and higher eGFR_{crea}. UACR and eGFR_{crea} can be high when there is less circulating serum creatinine (causing higher eGFR_{crea}) and, consequently, lower creatinine concentration in the urine (causing higher UACR values for fixed albumin concentrations). See also point 15 below.

Response: We thank the reviewer for raising this relevant question. We consider that correcting for creatinine is essential, due to variation in the general urine concentration depending on the urine volume. However, based on previous reports, the choice of the normalization method does not seem to have a major impact on the urinary metabolite levels, "... in general, the normalization method appears to have only minor influences on standard epidemiological regression analyses with clinical/physiological measures." (VP Mäkinen and M Ala-Korpela 2022).

To test if the lead metabolite signals are driven by the numerator (metabolite) or the denominator (creatinine), we have now investigated the association of the COJO lead variants with urinary creatinine concentration. None of the variants were associated with urinary creatinine at a Bonferroni corrected significance threshold of $p = 0.05/54 = 9 \times 10^{-4}$. This suggests that the observed signals are not driven by urinary creatinine. We have now added these results to a Supplementary Figure 4, and have commented on this issue in the results section:

"As we analyzed urinary metabolite to creatinine ratios instead of pure metabolite concentrations in order to account for the variation in the general urinary concentration and volume, we additionally tested whether the lead metabolite signals would be driven by the denominator (urinary creatinine) instead of the actual metabolite level (nominator). However, none of the COJO lead variants were significantly associated with urinary creatinine ($p > 0.05/54$ for all; Supplementary Figure 4), supporting that the observed signals are driven by the urinary metabolite levels".

2. After regressing the ratios on covariates, GWAS were conducted on the inverse normal transformation of the residuals. It is unclear if adjustment was also applied a second time, to prevent reintroduction of covariate effects (<https://www.nature.com/articles/s41431-018-0159-6>)

Response: We did not apply adjustment for covariates a second time for the INT residuals. We understand that this is a controversial topic, with many large genetic consortia using similar methods as we have done, i.e. adjusting only once for the covariates (e.g., Wuttke et al., Nat Genetics 2019/ CKDGen, PMID: 31152163). The main advantage of not performing additional covariate correction after the INT normalization is that it ensures normal distributions across all studies and phenotypes, avoiding skewed data.

To further investigate the effect of covariate associations, we tested the covariate associations with the transformed residuals in FinnDiane. Indeed, 17 of the metabolites showed a significant ($p < 0.05/54 = 9 \times 10^{-4}$) correlation with age, with median absolute correlation 0.11 ($Q_1 = 0.095$, $Q_3 = 0.14$). Furthermore, 25 metabolites were associated with sex (t-test $p < 0.05/54 = 9 \times 10^{-4}$) with median absolute difference of 0.24 ($Q_1 = 0.17$, $Q_3 = 0.34$).

However, other studies suggest that adjusting for environmental/ demographic factors such as age or sex can increase the statistical power, although adjusting for heritable/modifiable traits can indeed cause bias (Aschard et al, Am J Hum Genet 2015, PMID: 25640676). Since we only adjusted the models for non-modifiable risk factors, age and sex, instead of heritable/modifiable factors (such as BMI or eGFR), we would not assume that any of the identified variants would be primarily associated with these non-modifiable risk factors (age or sex), instead of the studied metabolites. Therefore, the covariate effect should not have any major impact on our analysis.

3. In the GWAS methods, it is described that “The association of genetic variants with urinary metabolites was tested using a frequentist test ...” Please, define the frequency test.

Response: We have now clarified that we used the score test as the frequentist test.

*“The association of genetic variants with urinary metabolites was tested with **score test with an additive model based on genotype probabilities to account for genotype uncertainty.**”*

4. It is unclear whether the genomic control correction was applied twice (at the individual study level and after the meta-analysis) or only before the meta-analysis. If meta-analysis results were not corrected, the genomic inflation factor of each meta-analysis should be reported in the supplement.

Response: The genomic control correction was applied only at the individual study level before the meta-analysis, and we have now clarified this in the methods:

“Meta-analysis of the individual GWAS was performed using METAL software (version 2011-03-25) applying inverse variance weighted method and genomic control correction for the individual study level results⁴⁷. The results were filtered to include variants with MAF ≥ 0.01 and found at least in 2 out of 3 studies. Genomic inflation factors λ_{GC} were calculated for the meta-analysis results (Supplementary Data 5).”

As suggested, we have now calculated the genomic inflation factor of each meta-analysis and have added the results as the Supplementary Data 5.

5. This study is based on meta-analyses of GWAS from very different studies: the FinnDiane on T1D patients and two general population studies from Scotland. The three studies have different phenotypic and covariate distributions, different genotype platform, etc. For these reasons, between-study heterogeneity statistics should be estimated, reported, and discussed in the case they affect any of the main findings and/or variants used as instrumental variables in Mendelian randomization (MR) analyses.

Response: Thank you for the suggestion. We agree that the three study populations have significant differences, especially due to the diabetes status affecting other covariate distributions. To address this potential heterogeneity, we had already calculated the between-study heterogeneity statistics for the heritability estimates, and we have now clarified the interpretation: 6 of the metabolites (cis-Aconitate, Histidine, Indoxyl Sulfate, Lactate, Tryptophan, and Valine) showed significant heterogeneity ($p < 0.05/54 = 9 \times 10^{-4}$), while none of the metabolites with significant heritability estimates based on the meta-analysis did. We have also clarified the corresponding paragraph in the results section:

“Altogether 27 of the 54 metabolites showed significant evidence of heritability ranging from 6% to 36% ($p < 0.05$), with highest estimates obtained for urinary citrate (36%, $p = 2.3 \times 10^{-20}$), 3-aminoisobutyrate (33%, $p = 1.7 \times 10^{-11}$), and tyrosine (29%, $p = 4.9 \times 10^{-12}$) concentrations (Supplementary Table 4, Supplementary Figure 1, Methods). Only six metabolites showed evidence of between study heterogeneity in the heritability estimates ($p < 0.05/54 = 9 \times 10^{-4}$), but none of the 27 metabolites with significant heritability estimates, supporting that there are no major differences in the genetic architecture of these metabolites between the studies.”

We have now in addition calculated the heterogeneity p -values for the GWAS meta-analysis lead signals (86 loci with COJO $p < 5 \times 10^{-8}$ to cover all variants used as IVs in the Mendelian Randomization) and have added them to the Supplementary Table 5 and Supplementary Table 14. Furthermore, we have added a forest plot with the study specific effect estimates for the urinary metabolite associations showing significant heterogeneity as Supplementary Figure 3.

We have accordingly added the following text to the results section:

“Nine of the 54 lead COJO signals showed evidence of between-study heterogeneity. These included 5 secondary signals for the 3-aminoisobutyrate locus on chromosome 5p13.2 in or near AGXT2, with the strongest associations obtained from the FinnDiane study with

individuals with T1D. Of note, for four of these secondary signals, the COJO estimate – conditional on other signals within the same locus – was in the opposite direction to the simple single-variant meta-analysis estimate, emphasizing the complexity of this locus with 13 detected independent signals (Supplementary Figure 3). Furthermore, the glycine association in the GM2A gene showed a trend in the opposite direction in the FinnDiane study compared to the two general population studies.”

Among the lead Mendelian Randomization findings, the trigonelline association at rs2106727 showed significant between-study heterogeneity. Therefore, the meta-analyzed effect size estimate, and thus also the causal effect size, is less precise. Nevertheless, the effect size estimates were trending in the same direction in all three cohorts, such that the causal association *per se* should persist.

We have added the heterogeneity as a limitation of the study to the discussion:

“The participants from the three included studies had also different clinical profiles with one of the studies including only individuals with T1D; notably, the mean age in the T1D cohort was 37.7 years vs 49.8 and 55.8 in the two general population cohorts. While the results were mainly concordant across the studies, we found evidence of heterogeneity in the heritability for six metabolites; furthermore, 9 of the 54 COJO lead signals demonstrated significant heterogeneity between the studies, with discrepant effect size directions in the T1D cohort for the glycine GM2A association and for 5 secondary 3-aminoisobutyrate signals in the chromosome 5p13.2 locus.”

6. If replication studies are available, the Authors may consider testing their novel findings for replication in independent studies (e.g.: Schlosser 2023, if applicable).

Response: Thank you for this suggestion. We had already searched for association signals from the GWAS catalogue for related traits to classify the identified associations as novel or previously known (i.e., previously reported corresponding urine/serum association in the GWAS catalogue), but following the reviewer’s suggestion, we have now performed a more comprehensive replication analysis for the novel lead loci (33 signals for 16 metabolites). We utilized the two following studies on urinary metabolites with genome-wide summary statistics available:

- Schlosser et al., 2023 (PMID: 37277652; including 11 of the 16 metabolites)
- Raffler et al., 2015 (PMID: 26352407; including 2 of the 5 missing metabolites and 9 of the 16 metabolites)

In addition, we considered a study by Calvo-Serra et al., 2021 (PMID: 33283231; urinary metabolites in children; including the remaining 3 missing metabolites and 9 of the 16 metabolites), however including only directly genotyped variants, and thus, did not cover our novel signals without replication.

We have now added the following text:

Methods: *“We tested replication of the 33 novel signals for 16 metabolites based on two previous urinary metabolite studies, with 11 out of the 16 metabolites found in a mass spectrometry-based study with 1,399 metabolites measured in 4,912 individuals⁷ and 9 out of the 16 metabolites found in an NMR-based study with 55 targeted metabolites measured in 5,552 individuals⁴. We found data for 24 of the 33 signals in either of the two studies. Summary statistics for Schlosser et al. 2023 were downloaded from the NHGRI-EBI GWAS Catalog⁴⁹ on 2024-04-24 and for Raffler et al. 2015 from <http://metabolomics.org/gwas> on 2024-05-13.”*

Results: *“We tested replication of the 33 novel signals for 16 metabolites based on two previous urinary metabolite studies^{4,7}. We found evidence of replication ($p < 0.05/24 = 0.0021$) for 10 of the 24 signals that were present in the replication data (Supplementary Data 1); in addition, 6 were significantly associated but in the opposite direction. Of note, 9 of the associations had a genome-wide significant p-value ($p < 5 \times 10^{-8}$) for replication, even though they were not reported in the GWAS catalogue; these were either due to i) a more stringent significance threshold required in the original study (e.g., rs2472479 3-hydroxyisobutyrate signal from Schlosser et al. 2023 study) ii) some of the older findings not being reported in the GWAS catalogue (e.g., rs62313082 associated with ethanalamine in Raffler et al. 2015); or iii) many of the secondary independent signals having been filtered out from the GWAS catalogue results (e.g. multiple of the 3-aminoisobutyrate signals on chromosome 5 and replicated in Schlosser et al. 2023).”*

7. As displayed in Suppl. Tab. 2, several metabolites are affected by a high proportion of missing values. Please, clarify if values are missing at random or if missingness is informative (eg: concentrations lower than the limits of detection). In the latter case, this would affect the distribution of the metabolites, reducing range and variability and data would be better imputed. This might also affect MR analyses involving the traits with largest share of missing data.

Response: The missing metabolite values indicate that the quantification was unsuccessful and missingness is, to the best of our knowledge, at random. Values below the limit of detection were set to the observed minimum for each metabolite, and we have now clarified this in the methods: *“In each study, the metabolite concentrations below the detection limit were set to the observed minimum for each metabolite.”*

The following sentence has been added to the discussion: *“We note that some of the metabolites had high missingness in some or all of the cohorts; to the best of our knowledge this missingness is random, however, it affected our power to detect both single variant and Mendelian Randomization associations and may hinder the metabolite applicability as biomarkers at least when using the current methods.”*

8. There is an emphasis on kidney function. Please, note that Wuttke et al. 2019 validated their findings on eGFR_{crea} against BUN, to identify a subset of loci that were more likely to pertain to kidney function rather than creatinine metabolism. Data on eGFR_{cys} are also available from the CKDGen Consortium. Those data may be relevant to clarify if the observed causal pathways belong to the kidney function or relate to creatinine metabolism (e.g.: discussion sentence “On the other hand, MR analysis suggested that urinary ethanolamine was associated with higher eGFR lending support for a potential causal protective role”).

Response: Thank you for the suggestion. We have now performed additional sensitivity analyses for the MR considering both blood urea nitrogen (BUN, Wuttke 2019) and cystatin C based eGFR (eGFR_{cys}, Stanzick 2021) data.

First, we used the Cystatine C based eGFR and the BUN data to perform sensitivity analyses on the MR with eGFR as the exposure and metabolites as the outcome. For this, we employed an approach similar to Wuttke et al. 2019, by creating two new eGFR instruments that included only IVs that were supported by BUN or eGFR_{cys} data to ensure relevance to kidney function:

eGFR&BUN: eGFR associated variants were used as instruments if BUN p-value < 0.05 and directionally consistent, i.e. effect for eGFR and BUN in opposite directions

eGFR&eGFR_{cys}: eGFR associated variants ($p < 5 \times 10^{-8}$) were used as instruments if eGFR_{cys} p-value < 0.05 and directionally consistent, i.e. eGFR and eGFR_{cys} in the same direction

In the original (updated) MR analysis eGFR was causally associated with 11 metabolites (ala, aohibut, bohival, ethh, form, gln, glya, leu, pseur, ura, val).

After pertaining only IVs supported by BUN or eGFR_{cys} data, eGFR&BUN was causally associated with 9 out of the 11, and eGFR&eGFR_{cys} with 10 out of the 11 metabolites ($p < 0.05 / 22$) and the direction of association was the same as in the original analysis. The following metabolites lost significance; eGFR did not affect 2-hydroxyisobutyrate and leucine after pertaining only IVs with reported effect on BUN, and pseudouridine after including only IVs with a reported effect on Cystatine C based eGFR. However, as all metabolites were causally affected by either eGFR&BUN or eGFR&eGFR_{cys}, we believe that these results further support that kidney function is indeed affecting the concentration of these metabolites in the urine. We have now added these results to the text in the results section:

“Finally, as some genetic loci associated with eGFR may reflect creatinine metabolism rather than kidney function, we performed sensitivity analyses by filtering the eGFR loci to those additionally associated ($p < 0.05$) with two alternative measurements of kidney function, namely cystatine C based eGFR²¹, and blood urea nitrogen³²; all detected eGFR-to-metabolite associations remained significant in one or both of these filtered analyses,

supporting that the observed causal pathways belong to the kidney function rather than relating to the creatinine metabolism (Supplementary Data 2).”

Next, we assessed whether the four metabolites (3-hydroxyhippurate, ethanolamine, quinic acid and trigonelline), implicated as causal for eGFR in our original updated MR, were also causal for BUN and eGFR-cys. Of these four, 3-hydroxyhippurate, quinic acid, and trigonelline were associated with BUN and cystatine C based eGFR, whereas ethanolamine was not after correction for multiple testing ($p < 0.05 / 8 = 0.0063$, Table below). Although the results for ethanolamine did not reach significance, ethanolamine was nominally associated with eGFRcys ($p=0.016$) and the direction of the BUN (-) and eGFRcys (+) matched the eGFR direction for ethanolamine, as well as for the other three metabolites. Furthermore, as none of the metabolite lead variants were associated with urinary creatinine (see response to question 1; $p > 0.05/54$ for all; Supplementary Figure 4), our results suggest that the ethanolamine association is not, at least solely, explained by association to creatinine metabolism. However, based on these results we have now modified the text accordingly, de-emphasizing ethanolamine:

Results: *“In addition, the genetic instrument for urinary ethanolamine, was associated with higher eGFR ($p=6.1 \times 10^{-8}$); of note, this association was not evident when using cystatin C based eGFR ($p=0.016$) and BUN ($p=0.18$) as alternative outcome measurements of kidney function (Supplementary Data 4).”*

Discussion: *“In the opposite direction, MR analysis suggested that urinary ethanolamine was associated with higher eGFR, potentially indicating a protective role. This finding, however, should be interpreted with caution as the association was not evident when using other measurements of kidney function, such as cystatine C based eGFR ($p=0.016$) or BUN ($p=0.18$).”*

Furthermore, we have added the necessary explanation to the methods section.

Supplementary Data 4. Results from two sample MR analysis: metabolite -> BUN and metabolite -> eGFRcys.

exposure	outcome	method	nsnp	b	se	pval	rsid
3-hydroxyhippurate	eGFR	Wald ratio	1	0.02	0.00	3.7×10^{-10}	rs6968554
3-hydroxyhippurate	BUN (Wuttke 2019)	Wald ratio	1	-0.04	0.010	1.6e-04	rs6968554
3-hydroxyhippurate	eGFRcys (Stanzick 2021)	Wald ratio	1	0.01	0.005	5.1e-03	rs6968554
Trigonelline	eGFR	Wald ratio	1	0.02	0.00	3.5×10^{-10}	rs2106727
Trigonelline	BUN (Wuttke 2019)	Wald ratio	1	-0.03	0.009	2.5e-04	rs2106727
Trigonelline	eGFRcys (Stanzick 2021)	Wald ratio	1	0.01	0.005	5.1e-03	rs2106727
Quinic acid	eGFR	Wald ratio	1	0.02	0.00	3.5×10^{-10}	rs2106727

Quinic acid	BUN (Wuttke 2019)	Wald ratio	1	-0.03	0.007	2.5e-04	rs2106727
Quinic acid	eGFRcys (Stanzick 2021)	Wald ratio	1	0.01	0.004	5.1e-03	rs2106727
Ethanolamine	eGFR	Wald ratio	1	0.01	0.00	6.1×10 ⁻⁰⁸	rs62313082
Ethanolamine	BUN (Wuttke 2019)	Wald ratio	1	-0.01	0.007	1.8e-01	rs62313082
Ethanolamine	eGFRcys (Stanzick 2021)	Wald ratio	1	0.01	0.004	1.6e-02	rs62313082

9. I might have missed the point, but I don't see a high value in presenting extensive heritability analyses, especially the between-cohort comparison, where heterogeneity is also provided. Genetic heritability is heterogenous across different contexts. In addition, with just 3 studies heterogeneity is typically underestimated and estimates not reliable.

Response: The meta-analysis of the heritability estimates across the three cohorts is our main outcome regarding the heritability estimates. As the reviewer points out, the three studies were different in their participant characteristics, and thus, the study-wise results and heterogeneity tests were performed as sensitivity analysis to identify any major differences. We agree that with three studies we are not fully powered to assess the heterogeneity, but the findings do support the conclusion, that there are no major differences in the genetic architecture between the cohorts impeding the meta-analysis. We have now modified the text to clarify the relevance of the heterogeneity analysis in the results section:

“Altogether 27 of the 54 metabolites showed significant evidence of heritability ranging from 6% to 36% ($p < 0.05$), with the highest estimates obtained for urinary citrate (36%, $p=2.3 \times 10^{-20}$), 3-aminoisobutyrate (33%, $p=1.7 \times 10^{-11}$), and tyrosine (29%, $p=4.9 \times 10^{-12}$) concentrations (Supplementary Table 4, Supplementary Figure 1, Methods). Only six metabolites showed evidence of between study heterogeneity in heritability estimates ($p < 0.05/54=9 \times 10^{-4}$), and none of the 27 metabolites with significant heritability estimates, supporting that there are no major differences in the genetic architecture of these metabolites between the studies.”

10. eQTL analysis covers an extensive part of the Results and becomes difficult to follow. To identify the most likely causal gene underlying detected signals, I would suggest trying to present eQTL analysis and colocalization with gene expression analyses in a more organized way, setting some rules of thumb based on clear cutoffs to identify the most likely causal gene.

Response: Thank you for this suggestion. We have accordingly reformatted the whole eQTL analysis results section, with this feedback in mind. The original text was quite detailed, and we understand that it was hard to follow. We have therefore now both reorganized and rewritten the text to make it clearer.

However, it is challenging to set rules for defining the causal genes; sometimes the kidney eQTL seems to provide the most plausible gene, whereas in other cases the blood eQTL/colocalization provides the more logical candidate. We think this discrepancy may be due to tissue specific expression patterns for some of the genes. In some cases, neither of the eQTL datasets provided the most plausible gene, instead the most logical causal gene seemed to be the closest gene as in the case of the 1-Methylnicotinamide-associated variant within the *ACSMD* gene encoding for alpha-Amino-beta-carboxymuconate-epsilon-semialdehyde decarboxylase, the key enzyme regulating de novo synthesis of nicotinamide adenine dinucleotide. To reflect this, we have added a new column to the eQTL table indicating the closest gene and other relevant nearby genes and simplified the table.

11. It is not clear what the purpose of this analysis was: “As eGFR causally affected multiple metabolites we tested if adjusting the GWAS analysis with eGFR affected the metabolite associations in FinnDiane. However, adjustment with eGFR had little effect on the effect estimate for the 41 COJO lead variants included in the FinnDiane GWAS data set (Suppl. Tab. 7).”

Response: Since our analysis suggested that eGFR causally affects urinary concentrations of multiple metabolites, we hypothesized that some of the lead metabolite signals could be related to kidney function (eGFR). The purpose of the eGFR adjusted analysis was to assess whether kidney function had an effect on the observed associations. We have now clarified this part of the text in the results section:

*“As eGFR causally affected multiple metabolites, we tested **whether some of the metabolite GWAS associations might be driven by association with eGFR**. However, **adjusting the analysis with eGFR in the FinnDiane study did not markedly affect the effect size estimates for the 41 COJO lead variants included in the FinnDiane GWAS data set (Supplementary Figure 9)**.”*

12. For MR analyses, instrumental variables should be selected if they have an F statistic ≥ 10 .

Response: We have now estimated the F statistic for each instrumental variable (all metabolites, eGFR, eGFR/BUN, UACR, and BMI) accordingly. The F statistics for all IVs for eGFR, eGFR/BUN, UACR, and BMI, had an $F > 10$. Altogether 82 of the IVs for the metabolites had an $F > 10$. According to the reviewer’s feedback, we have now filtered out any IVs with an $F < 10$ to ensure a strong instrument (4 filtered). We have also updated our MR analyses to reflect these changes. The updated MR results are in line with the original results.

13. To support the MR analyses, evidence that the selected instrumental variables satisfy the key assumptions is needed. This is especially important for the IVs used for the metabolites, as

they may be influenced by lifestyle and thus violate the assumption of no hidden confounding between IV and exposure. How the Authors excluded the presence of horizontal pleiotropy is also important information.

Response: Thank you; we agree that ensuring the robustness of Mendelian Randomization requires assessment of the underlying assumptions, and we have now stated these more clearly. To assess that our genetic instruments are strongly associated with the exposures (metabolites or eGFR), we only included genetic variants as IVs that were genome-widely associated ($p < 5 \times 10^{-8}$) with the traits (MR relevance assumption). In addition, we have also now calculated the F statistics for each SNV-association as an additional verification of this assumption. We have highlighted this in the methods section; Mendelian randomization:

Methods (Mendelian Randomization): *“To ensure that the instruments are strongly associated with each respective exposure (relevance assumption), we selected only genetic variants as instruments that showed a strong and independent association with the exposure. Therefore, we included only variants that were genome-widely and independently associated with the exposure ($r^2 = 0.001$, $p < 5 \times 10^{-8}$). Furthermore, for each genetic variant we estimated the F-value using the formula $F = \beta^2 / \text{standard error}^2$, and included only IVs with and $F \geq 10$.”*

To further test the presence of pleiotropy that relates to the second and third assumption (independence and exclusion restriction criteria assumptions), we assessed causality using methods that are less sensitive to pleiotropy or make different assumptions regarding pleiotropy. These methods allow half (weighted median), a subset (simple and weighted mode) or all of the variants (MR Egger) to be pleiotropic. If the effect estimates from these methods agree, we determined that pleiotropy is less likely to have influenced our results. We also tested the heterogeneity in our MR associations, which we have now clarified in the text (we also quantified the heterogeneity with the Inconsistency index). In addition, we added the intercept test as a sensitivity analysis (see response below to comment 14), which also aid the detection of potential pleiotropy. We have now added all changes to the methods section:

Methods (Mendelian Randomization): *“For exposures with 3 or more IVs, causality was further assessed using methods less sensitive to pleiotropy. These methods assume that only a subset of the variants are valid instruments (i.e., not pleiotropic), including weighted median, simple and weighted mode MR methods. We also used MR-Egger, which allows all variants to have pleiotropic effects, under the InSIDE (Instrument Strength Independent of Direct Effect) assumption. We additionally tested if the Egger intercept deviated from zero with the Egger intercept test, as this may indicate either horizontal pleiotropy or a violation of the InSIDE assumption underlying MR-Egger⁶². For MR associations with multiple IVs, we used the Cochran Q test to examine any heterogeneity of SNVs’ estimates.”*

We have also slightly modified the results section to better reflect the fact that we also considered the MR assumptions:

Results (MR: Kidney health causally affects urinary metabolites): *“To avoid violation of the relevance assumption underlying MR, we used a genetic instrument for eGFR, composed of 150 independent ($r^2=0.001$) genome-wide significant SNVs identified in the CKDGen GWAS meta-analysis¹⁷”*

Results (MR: Kidney health causally affects urinary metabolites): *“Next, we used complementary MR methods (median- and mode-based methods as well as MR Egger), with differing underlying assumptions, to assess the robustness of our findings with respect to pleiotropy. The causal estimates were directionally consistent across different MR analysis-methods, suggesting no significant pleiotropy for most of the metabolites. Moderate heterogeneity between the variant-specific causal estimates (30-32%, Supplementary Table 9), were detected for valine and alanine. However, the Egger intercept did not significantly deviate from zero ($p>0.05$, Supplementary Table 9)”*

These complementary sensitivity analyses were however only possible for IVs with 3 or more genetic variants. Many of our metabolites were affected by only one or two variants, preventing implementation of such testing. However, we also performed an extensive look-up in the GWAS catalogue of all the urinary metabolite-associated variants (including variants in LD, $r^2>0.80$), and highlighted pleiotropic effects noted in the text. E.g., two of our genetic instruments (rs6968554; IV for 3-hydroxyhippurate rs2106727; IV for Quinic acid Trigonelline), were both in within the *AHR* gene locus (known for its strong connection with coffee intake) and previously strongly associated with coffee consumption /caffeine intake (more details in response 15 and 17 below). The genetic instrument for ethanolamine (rs62313082), had not previously been reported in the GWAS catalogue, which has now been clarified in the text in the results section along with the look-up for the remaining genetic instruments:

Results (MR: Urinary metabolites potentially causally linked to kidney function and body mass index): *“The genetic instrument was based on the rs62313082 variant which is also a kidney eQTL for the ETNPPL gene (Table 2 and 4) but has not been associated with other traits in the GWAS catalogue (Supplementary Table 6). Moreover, the MR analysis suggested that the genetic instruments for urinary 1-methylnicotinamide ($p=2.3\times 10^{-5}$) and 4-deoxythreonate ($p=7.6\times 10^{-5}$) were associated with higher body mass index (BMI; Table 4). These were genetically instrumented by rs78470967 and rs181558 (4-deoxythreonate), and rs17322446 (1-methylnicotinamide), for which we found a few associations in the GWAS catalogue (rs17322446; Quinolate levels, walking pace and FEV1, rs181558; type 2 diabetes and height, Supplementary Table 6).”*

However, we acknowledge the challenge to distinguish vertically pleiotropic associations (does not violate MR assumption) from horizontally association (violates MR assumption) by only a look up in the GWAS catalogue.

14. For use of the MR-Egger, are the INSIDE assumptions met?

<https://journals.plos.org/plosgenetics/article?id=10.1371/journal.pgen.1010166>

Response: Thank you for the comment. We have now conducted additional sensitivity analyses, such as the Egger intercept test, to verify the InSIDE (Instrument Strength Independent of Direct Effect) assumption. Our findings indicate that the Egger intercepts did not significantly deviate from 0 (see table below). This suggests that, on average, there is no substantial directional pleiotropy (Burgess et al 2017, PMID: 28527048), implying either the absence of directional pleiotropy or that the INSIDE assumption holds. However, we acknowledge the difficulty of truly testing the InSIDE assumption. We have now added these to the methods and results section:

Methods (Mendelian Randomization): *“We also used MR-Egger, which allows all variants to have pleiotropic effects, under the InSIDE (Instrument Strength Independent of Direct Effect) assumption. We additionally tested if the Egger intercept deviated from zero with the Egger intercept test, as this may indicate either horizontal pleiotropy or a violation of the InSIDE assumption underlying MR-Egger⁶².*

Results (MR: Kidney health causally affects urinary metabolites): *There was no indication that the genetic instruments showed horizontal pleiotropy, as indicated by the Egger intercept test (p>0.05, Supplementary Table 9).*

Outcome	Method	Beta (SE)	P	P _{het}	I ² (%)	Egger intercept (p)
Glycolic acid	MR Egger	2.76 (1.24)	2.7×10⁻⁰²	2.0×10 ⁻⁰¹	9.13	0.00 (0.40)
Ethanolamine	MR Egger	0.69 (1.08)	5.2×10 ⁻⁰¹	5.2×10 ⁻⁰¹	0	0.00 (0.89)
Valine	MR Egger	0.93 (1.28)	4.7×10 ⁻⁰¹	8.0×10 ⁻⁰⁴	29.51	0.01 (0.16)
Uracil	MR Egger	2.06 (1.05)	5.2×10 ⁻⁰²	3.9×10 ⁻⁰¹	2.77	0.00 (0.98)
Formate	MR Egger	1.84 (1.12)	1.0×10 ⁻⁰¹	1.4×10 ⁻⁰¹	11.56	0.00 (0.78)
Leucine	MR Egger	0.50 (1.11)	6.5×10 ⁻⁰¹	1.4×10 ⁻⁰¹	11.60	0.01 (0.12)
Glutamine	MR Egger	1.70 (1.21)	1.6×10 ⁻⁰¹	1.3×10 ⁻⁰¹	12.13	0.00 (0.79)
2-Hydroxyisobutyrate	MR Egger	1.58 (1.05)	1.4×10 ⁻⁰¹	4.2×10 ⁻⁰¹	2.01	0.00 (0.88)
Alanine	MR Egger	0.81 (1.28)	5.3×10 ⁻⁰¹	2.6×10 ⁻⁰⁴	31.78	0.01 (0.27)
3-Hydroxyisovalerate	MR Egger	3.40 (1.24)	6.7×10 ⁻⁰³	3.3×10 ⁻⁰³	33.71	-0,01 (0.15)
Pseudouridine	MR Egger	2.20 (1.10)	4.6×10 ⁻⁰²	2.2×10 ⁻⁰²	20.22	0.00 (0.58)
Pyroglutamate	MR Egger	0.98 (1.06)	3.6×10 ⁻⁰¹	6.8×10 ⁻⁰¹	0.00	0.00 (0.54)
Glycine	MR Egger	-0.67 (1.71)	7.0×10 ⁻⁰¹	1.6×10 ⁻²¹	61.16	0.01 (0.16)

15. The Authors observed that higher urinary 3-hydroxyhippurate, quinic acid, and trigonelline concentrations are causally associated with higher BMI, lower urinary creatinine concentration, higher UACR and higher eGFR estimated from serum creatinine. Given higher levels of the 3 metabolites cause lower urinary creatinine, if urinary albumin is not altered, this would cause

higher UACR. And if the 3 metabolites lower urinary creatinine through lowering serum creatinine, that would be compatible also with higher eGFR_{crea}. All of this does not necessarily implicate kidney health.

Response: Thank you for the comment. We have now revised the paragraph to clarify our interpretation of the results. In particular, with “reflecting better/worse kidney health”, we refer to eGFR and UACR, (i.e., not to the three metabolites), to help the readers who may not be familiar with the various measurements of kidney parameters. We have also removed altogether the analysis of urinary creatinine measurements, due to the difficulties in the clinical interpretation of the trait (reflecting at the same time e.g. urine volume).

“The analysis suggested that higher urinary 3-hydroxyhippurate, quinic acid and trigonelline concentrations are causally associated with higher body mass index (BMI), higher eGFR (i.e., reflecting better kidney health), and contradictorily, also with higher UACR (i.e., reflecting worse kidney health; Table 4). All these three metabolites are found in coffee; their instrumental variables (IVs) rs2106727 and rs6968554 are located in the AHR locus and are in strong LD with rs4410790 that has been previously associated with caffeine intake ($p=2.0 \times 10^{-249}$)³⁴, suggesting that the metabolite associations observed with Mendelian Randomization may reflect the underlying association with coffee consumption. Indeed, in line with our observation of the three metabolites associated with higher eGFR, a previous Mendelian Randomization study suggested that coffee consumption has a beneficial effect on kidney function and albuminuria³⁵. Why the three urinary metabolites were simultaneously associated with higher UACR in our data remains unclear. For BMI, previous MR studies have found contradictory evidence regarding the causality between coffee consumption and BMI or obesity^{36,37}. One challenge may be the reverse causality; in our study, BMI was not causally affecting the three metabolites, or any other urinary metabolite (Supplementary Data 3). In general, the urinary metabolites may provide a more exact estimate of the coffee intake than self-reported data on coffee consumption, and thus, our findings add to the previous evidence of a BMI-increasing effect on coffee consumption.”

16. About the causal effect on BMI, checking for reverse causation would be good, as BMI might play the role of a confounder (see an example in PMID 31306056). On the other hand, given the causal effect of BMI on eGFR (direct) and UACR (mediated by DM) (PMID 36305100), multivariate MR of the 3 metabolites on kidney health through BMI would be informative.

Response: Thank you for this insight; we have now run a Mendelian Randomization testing whether BMI is causal for our urinary metabolites, and added a description of this in the Methods:

“For BMI, we used 458 variants associated with BMI (439 independent variants after clumping, 419 after effect allele harmonizing) in UK biobank available in the manually curated OpenGWAS database (Dataset: ukb-b-19953)”

The results suggested that BMI does not causally associate with the urinary metabolites in our study (added as a Supplementary Data 3). Since we observed no effect of BMI on the metabolites (in contrast to PMID 31306056 where the authors observed strong associations both ways), the multivariable MR will likely not provide any new information, although we agree this would have been an interesting way to study the relationships if BMI would have turned out to be causal.

17. Finally, the Authors note that all 3 metabolites are found in coffee. Coffee consumption is related to human behavior and is typically related to other social factors such as education, alcohol consumption, smoking. In this case, it is recommended to exclude violation of MR assumptions for all IVs related to the three metabolites.

Response: Yes, although we agree that coffee consumption is related to human behavior, which may violate the MR assumptions, the MR analysis with these metabolites (urinary 3-hydroxyhippurate, quinic acid and trigonelline) as exposures were genetically instrumented by only 1-2 genetic variants. Therefore, excluding the IVs due to their association with coffee consumption (and thus hypothetically human behavior) would mean excluding the whole MR for that metabolite.

However, in our extensive look-up in the GWAS catalogue (Supplementary Table 6), we did not observe any associations with smoking or educational attainment for the IVs used for these three metabolites (rs2106727, rs6968554), but instead they were associated with the amount of coffee and caffeine in multiple GWAS. As the metabolites are found in coffee and affected by SNVs that are both biological candidates and strongly associated with the amount of caffeine/coffee intake, they may represent vertically pleiotropic associations (which do not violate the third MR assumption). In other words, it may be that the metabolite is on the pathway; coffee consumption → increase levels of urinary 3-hydroxyhippurate, quinic acid and trigonelline → kidney health; or alternatively, these metabolites may serve as a proxy/biomarker for caffeine intake.

Minor issues:

18. Introduction: compared to e.g. Schlosser et al 2023, I wouldn't say that 54 metabolites correspond to an "extensive molecular coverage"

Response: We agree. We have now clarified the sentence to suggest that the NMR platform provides a compromise between the individual urine laboratory measurements and the mass spectrometry-based, more extensive experiments, as well as added this to the discussion paragraph on the study limitations:

Introduction: *"Here, balancing between large sample size and molecular coverage, we have utilized the urinary NMR metabolomics platform detecting 54 urinary metabolites in 8,026 individuals to further characterize the genetics of urinary metabolites."*

Discussion: *“Finally, the urinary NMR platform that we utilized was able to detect 54 urine metabolites; while this is significantly more than e.g., the genetic analyses performed for the four urinary laboratory measurements in the UK Biobank⁸, the number of metabolites that can be captured with the NMR platforms remains far below the width of metabolites detected e.g., with mass spectrometry based methods⁷.”*

19. Introduction: it is not a prerequisite of MR that the exposure and the outcome should be associated to test for a causal effect. Sometimes, the association cannot be seen for the presence of confounders. Maybe, list the three basic hypotheses of MR, which should hold for valid inference.

Response: We agree with the reviewer, and have now clarified that in the text:

Introduction (second last paragraph): *“Studying the genetic factors associated with urinary metabolites has also other benefits beyond explaining the underlying biological mechanisms. In particular, if genetic variants can be identified for a urinary metabolite, these variants can serve as genetic instruments in Mendelian randomization (MR) to infer causality between the metabolite and disease outcomes. The identification of multiple robust variants provides more reliable MR results, i.e., by enabling the use of MR methods that account for pleiotropic effects. Although our study includes fewer metabolites than many mass spectrometry-based studies, the larger sample size gives us power to potentially identify more associations with urinary metabolites.”*

In addition, we have now shortly listed the three basic hypotheses of MR in the results section:

“MR assumes that the genetic variants (instrumental variables) are associated with the exposure, are not associated with any confounders of the exposure-outcome relationship and impact the outcome only through the exposure and not via independent pleiotropic pathways.”

20. Introduction: the 4th paragraph reads more like a discussion than an introduction.

Response: We have now reformatted the paragraph to better match the introduction (see above).

21. Results: what to the Authors mean with “study-wide”?

Response: The “study-wide significance” was defined as the genome-significance threshold ($p < 5 \times 10^{-8}$) Bonferroni-corrected for 54 studied metabolites, $p < 9.3 \times 10^{-10}$. We have now removed “study-wide”, and simply use “significant ($p < 9.3 \times 10^{-10}$)”.

22. Results: the citrate-associated rs11567842 was tested for association with several traits. I think that a P-value of 0.012 with eGFRcrea from a meta-analysis of >700,000 individuals is not really meaningful.

Response: We agree with the reviewer and have now removed the part of the sentence referencing the general population eGFR.

23. I am not sure I fully agree with sentence: “However, all significant MR analysis findings were based on only one or two significant variants available for each metabolite and need to be interpreted with caution; the largest number - 8 genetic variants – were available for MR for 3-hydroxyisovalerate, which was not associated with any of the studied outcomes ($p>0.01$)”. Power to detect a causal association also depends on the strength of the IV and their specificity. For molecular data we are often in the presence of a single variant with large effect on the exposure.

Response: Yes, we agree with the reviewer that the number of variants does not solely determine the power. The actual message we wanted to convey was that we could not perform any formal testing of pleiotropy as only 1-2 IVs were available. We have now modified this sentence accordingly:

“Of note, as these MR findings were based on only one or two genetic variants, complementary MR methods more robust to pleiotropy could not be implemented.”

24. Discussion: I didn’t understand the sentence: “This finding suggests that the glomerular filtration rate needs to be considered when investigating these metabolites as potential biomarkers of disease risk”. Can the Authors try to rephrase?

Response: We have now clarified the sentence:

*“This finding suggests that the **clinical studies** investigating these metabolites as potential biomarkers of disease risk **should include the glomerular filtration rate as a covariate in the analysis.**”*

25. Study limitation section can be expanded.

Response: Thank you for the suggestion. We have now expanded the limitation section in the discussion:

*“As a limitation, the study participants were mostly of European origin and further studies are required to investigate the generalizability of our findings to other populations. **The participants from the three included studies had also different clinical profiles with one of the studies including only individuals with T1D; notably, the mean age in the T1D cohort***

was 37.7 years vs 49.8 and 55.8 in the two general population cohorts. While the results were mainly concordant across the studies, we saw evidence of heterogeneity in the heritability of six metabolites; furthermore, 9 of the 54 COJO lead signals demonstrated significant heterogeneity between the studies, with discrepant effect size directions in the T1D cohort for the glycine GM2A association and for 5 secondary 3-aminoisobutyrate signals in the chromosome 5p13.2 locus. We note that some of the metabolites had high missingness in some or all of the cohorts; while this missingness is random to the best of our knowledge, it affected our power to detect both single variant and Mendelian Randomization associations, and may hinder the metabolite applicability as biomarkers at least when using the current methods. Finally, the urinary NMR platform that we utilized was able to detect 54 urine metabolites; while this is significantly more than e.g., the genetic analyses performed for the four urinary laboratory measurements in the UK Biobank,⁸ the number of metabolites that can be captured with the NMR platforms remains far beyond the width of metabolites detected e.g., with mass spectrometry-based methods.⁷”

26. Methods: pleiotropy and heterogeneity are two different things; there might be situations where heterogeneity does not implicate pleiotropy.

Response: We agree. We have now clarified this by removing the word “heterogeneity”.

“For exposures with 3 or more IVs, causality was further assessed using methods less sensitive to pleiotropy.”

Reviewer #2 (Remarks to the Author):

The present study aimed to identify genetic influences on the concentrations of 54 urinary metabolites and evaluate their impact on renal function using genome-wide association studies (GWAS) across three European cohorts, meta-analyses, and Mendelian randomization analysis. Specifically, the authors identified 26 chromosomal regions associated with at least one of the 54 metabolites analyzed. They identified 52 associations in 19 of the 54 studied metabolites, of which 31 were novel. These variants, primarily located in regulatory regions, show a strong association with gene expression in renal tissue, tubules, and glomeruli. Furthermore, subsequent Mendelian randomization analysis suggests that the estimated glomerular filtration rate (eGFR) causally influences 13 urinary metabolites, and there is an association between urinary ethanolamine and high eGFR levels, suggesting a potential protective role. This association is based on rs62313082, which correlates with higher urinary ethanolamine concentrations, higher eGFR, and lower expression of the ETNPPL gene in the kidneys.

This study provides a starting point for future analyses to support experimental validation or biological hypotheses originating from observational studies.

1. Overall, the article presents a highly complex and well-articulated analysis in all its parts, with interesting and well-exposed results; however, clarifying the workflow of the entire work would be beneficial.

Response: We thank the reviewer for the overall positive feedback! We agree that the study design includes multiple subanalyses, and have now clarified the main study workflow in the introduction, and site the Figure 1 with a schematic illustration already in this paragraph:

“In this study, we performed genome-wide association study (GWAS) meta-analyses in a total of 8011 individuals to investigate single nucleotide variants (SNVs) associated with 54 urinary metabolites measured by NMR in one Finnish cohort of individuals with type 1 diabetes (T1D) and two Scottish cohorts from a general population setting. After conditional analysis to identify independent secondary signals, we characterized the identified associations and their molecular basis by analysing the variants’ effect on gene expression harnessing relevant expression quantitative trait loci (eQTL) data, and performed pathway analyses to obtain wider understanding of the biological pathways affecting each metabolite. Finally, we assessed causal relationships between the metabolites and relevant phenotypes and health outcomes using MR analysis (Figure 1).”

In addition, we have extended the Figure 1 legend to better describe the workflow of the entire work.

2. In the results section, the total number of individuals analyzed (8026) does not match the sum of the numbers of individuals in the three cohorts reported in the first paragraph of the results and in Supplementary Table 1. Additionally, in Figure 1, the number of individuals is incorrect (the number of individuals reported in the VIKING cohort is 2077 instead of 2027 compared to the previous data).

Response: We thank the reviewer for noting these errors. The number of individuals analyzed from the VIKING cohort should be 2024, not 2027, and the total number of individuals analyzed should be 8011, not 8026. We have now updated the manuscript throughout.

3. In the Methods section, I appreciate the detailed description of the cohorts examined. However, for a more accurate evaluation, it would be advantageous to clearly understand the exclusion criteria applied to each cohort. Therefore, specifying the exclusion criteria used, including the actual number of excluded individuals rather than providing approximations, would be beneficial.

Response: Thank you for the suggestion. We have now added flow charts for each of the three studies to describe the main exclusion criteria used to include/exclude individuals (Supplementary Figure 10).

Methods: “*Urinary metabolomics by NMR was measured for all available and qualifying FinnDiane and VIKING study participants, and for 2,743 of the ~20,000 Generation Scotland participants with urine sample available (Supplementary Figure 10).*”

Supplementary Figure 10: Participant selection criteria in FinnDiane, VIKING and Generation Scotland studies. ESKD: end-stage kidney disease. T1D: type 1 diabetes.

4. From Supplementary Table 2, it is observed that some metabolites have a high number of missing readings, with different percentages between the two cohorts (e.g., mannitol and taurine). Has this kind of issue been addressed in any way?

Response: The differences in missingness between the cohorts have not been directly addressed in the analysis. To the best of our knowledge, however, the missingness was at random, and the differences in missingness between the cohorts should not bias the results. We note that we studied the heterogeneity in the heritability estimates across the three studies, and only 6 of the 54 metabolites showed evidence of heterogeneity between the studies ($p < 0.05/54$: cis-aconitate, histidine, indoxyl sulfate, lactate, tryptophan, and valine), with no heterogeneity for e.g. mannitol and taurine ($p_{het} > 0.05$).

However, we acknowledge that we were not able to obtain significant heritability estimates for many of the metabolites with high missingness values across the cohorts (e.g., 3-Methylhistidine, Creatine, histidine, isoleucine, Propylene Glycol, and taurine). We have now clarified this in the discussion:

Discussion: *“We note that some of the metabolites had high missingness in some or all of the cohorts; while this missingness is random to the best of our knowledge, it affected our power to detect both single variant and Mendelian Randomization associations, and may hinder the metabolite applicability as biomarkers at least when using the current methods.”*

5. Two notes: In line 47, the number of previously unreported associations was 31, not 32. In lines 623, 642, and 645 the GCTA tool is referred to as the GTCA .

Response: Thank you for noting these. The correct number of novel associations is 33 and has been corrected to the manuscript.

We have updated the COJO analysis by excluding first and second-degree relatives from the reference sample. This resulted mostly in minor changes in the effect estimates and p-values for loci with multiple independent signals for the same metabolite. More notably, the number of independent signals for 3-aminoisobutyrate on chromosome 5 increased from 13 to 15 (3 old signals were excluded and 5 new signals were included) resulting in a total of 33 novel signals instead of 31.

REVIEWER COMMENTS

Reviewer #1 (Remarks to the Author):

I would like to thank the Authors for considering all points raised, recognizing that they implicated substantial amount of work. I still one major comment (the first one) and a few minor issues (the remainder) to highlight:

Main issues

1. I appreciate the response to my previous point #2, but I think the evidence that the residuals of 17 out of 54 metabolites were still associated with age and sex confirms the alert by Pain et al (<https://www.nature.com/articles/s41431-018-0159-6>) that normalization may reintroduce the covariate effect. I agree that models should not be adjusted for modifiable factors, but it would be important to demonstrate that age and sex had no role in determining the observed associations, especially given downstream causal analyses involved kidney function, which is related to age and sex.

Response: We agree with the reviewer that it is important to demonstrate that age and sex are not driving the observed associations due to re-introduced covariate effects. To this end we have now re-analyzed the association of the COJO lead signals (n=54) with urinary metabolites in the FinnDiane study with an additional model. The original analysis was performed by regressing the metabolite values on the covariates (age, sex, genetic principal components, and genotyping batch) followed by inverse normal transformation (INT) of the residuals (MIN model). The INT residuals were used as the outcome in the genetic association analysis. We have now performed genetic association analysis further adjusting for age and sex to prevent the re-introduction of covariate effects (MIN model + age and sex).

The association effect estimates were very similar to the original analysis (Pearson correlation coefficient = 0.9996) demonstrating that age and sex had no major role in determining the observed associations. We have added Supplementary Figure 9B (see below) plotting the original effect estimates (MIN model) against the effect estimates from the updated model (MIN model + age and sex).

We have also updated the effect of eGFR adjustment analysis since it included the old COJO lead signals and excluded variants with imputation quality < 0.7.

The following text has been added to the methods:

“The effect of different covariate adjustment strategies were tested in FinnDiane for the COJO lead signals by analysing two additional models: 1) eGFR was added as a covariate before the INT of the residuals prior to the genetic association analysis (Supplementary Figure 9A), and 2) age and sex were included as covariates in the genetic association analysis to test if age and sex are affecting the observed metabolite associations by re-introduced covariate effects after the INT of the residuals (Supplementary Figure 9B)⁵⁴.”

2. Regarding my previous point #6, significant association in the opposite direction does not constitute replication. Replication should be evaluated with one-sided tests. Maybe it should be commented why association of the same variant with the same metabolite can be significant in the opposite direction.

Response: We agree that significant association in the opposite direction does not constitute replication. We were unable to evaluate replication with one-sided tests since replication was performed as look-ups from existing GWAS summary statistics with two-sided test results.

We investigated possible reasons why the association in the discovery and replication could be in the opposite direction and noticed a mistake in the code that compiles the replication data table (Supplementary Data 1). The effect allele and non-effect allele were flipped in the replication study, if the effect allele in the replication study corresponded to the non-effect allele in the metabolite GWAS. However, the sign of the corresponding effect estimate in the replication data set was by mistake not updated. After correctly updating the sign of the effect estimates in the replication data, none of the significant associations were in the opposite direction.

We have updated Supplementary Data 1 and modified the paragraph on replication results accordingly:

“We found evidence of replication ($p < 0.05/24 = 0.0021$ and concordant effect direction) for 16 of the 24 signals that were present in the replication data (Supplementary Data 1).”

3. Regarding my previous point #7, I agree with the solution chosen by the Authors to impute missing data to the limit of detection but please, realize that this contradicts the assumption of “missingness at random”: in fact, missingness is informative and missing data can reliably be assumed to represent low concentrations. I recommend revising the comment in the manuscript.

Response: We agree that in the case presented by the reviewer the missingness would be informative. However, the missing values were not imputed but were left missing. Instead, we have only imputed the values below the limit of detection to minimum observed metabolite value for each metabolite. We have now clarified this in the methods:

“In each study, the metabolite concentrations below the detection limit were set to the observed minimum for each metabolite and missing values were left missing.”

4. Great that the Authors analyzed urinary creatinine separately. Suppl. Fig. 4 shows that for most metabolites, the association with creatinine was nearly null, supporting that for most of them the signals are driven by the metabolite levels. However, despite not being significant after multiple-testing correction, for a couple of them the effect on creatinine is not negligible (eg the lowest blue point, maybe 3-Hydroxyisovalerate? Difficult to recognize by the legend). Consequently, I would suggest using a less definitive wording in the comment, by saying eg that supporting that the observed signals are *generally/mostly* driven by the urinary metabolite levels (or similar).

Response: We agree with the suggestion and have revised the text that now reads:

“However, none of the COJO lead variants were significantly associated with urinary creatinine ($p > 0.05/54$ for all; Supplementary Figure 4), supporting that the observed signals are mostly driven by the urinary metabolite levels.”

5. The claimed number of metabolites is 54 but conclusions (paper and abstract) refer to 53, please clarify

Response: We quantified 54 metabolites but the GWAS results for glucose were spurious in each of the three cohorts and are not reported. Consequently, glucose was excluded from the downstream analysis. To clarify this, we have added a sentence to the results section:

“The GWAS results for glucose were spurious and hard to interpret, and are therefore not reported.”

6. Suppl. Tab. 6: what does the yellow highlight represent?

Response: The yellow highlight was to mark one of the previously reported associations for the same metabolite to help identify previously reported signals. We have now removed the highlight from the Supplementary Table 6.

7. Suppl. Tab. 11: what does boldface of last column items represent?

Response: The boldface indicates that the gene is nominally associated with the metabolite in the gene level analysis. We have moved the sentence “Genes in ***bold cursive*** were nominally associated with the metabolite ($p < 0.05$) in the MAGMA gene level analysis.” from the table footer to the table caption to make it clearer.

8. Suppl. Tab. 12: please, fix a typo in the title (“significant”)

Response: Thank you for noting this. We have now corrected the spelling.

Reviewer #2 (Remarks to the Author):

1. The authors responded properly and completely to the reviewers' comments. However, in the manuscript, in lines 707, 710 and 751, the name of the GTCA tool is incorrect. Please correct it to GCTA.

Response: Thank you for pointing this out. We have now corrected this mistake in the revised manuscript.